# Visualization and design of the functional group distribution during statistical copolymerization

Paul H.M. Van Steenberge [1], Ondrej Sedlacek [2], Julio C. Hernández-Ortiz[1], Bart Verbraeken[2], Marie-Françoise Reyniers[1], Richard Hoogenboom [2] & Dagmar R. D'hooge [1,3]

Even though functional copolymers with a low percentage of functional comonomer units (up to 20 mol%) are widely used, for instance for the development of polymer therapeutics and hydrogels, insights in the functional group distribution over the actual chains are lacking and the average composition is conventionally used to describe the functionalization degree. Here we report the visualization of the monomer distribution over the different polymer chains by a synergetic combination of experimental and theoretical analysis aiming at the construction of functionality-chain length distributions (FUNC-CLDs). A successful design of the chemical structure of the comonomer pair, the initial functional comonomer amount (13 mol%), and the temperature (100 °C) is performed to tune the FUNC-CLD of copoly(2-oxazoline)s toward high functionalization degree for both low (100) and high (400) target degrees of polymerization. The proposed research strategy is generic and extendable to a broad range of copolymerization chemistries, including reversible deactivation radical polymerization.

[1] Ghent University, Laboratory for Chemical Technology (LCT), Technologiepark 125, B-9052 Gent, Belgium. [2] Ghent University, Supramolecular Chemistry Group, Centre of Macromolecular Chemistry (CMaC), Department of Organic and Macromolecular Chemistry, Krijgslaan 281-S4, 9000 Gent, Belgium. [3] Ghent University, Centre for Textile Science and Engineering, Technologiepark 70a, B-9052 Gent, Belgium. Correspondence and requests for materials should be addressed to R.H. (email: Richard.Hoogenboom@UGent.be) or to D.R.D. (email: Dagmar.Dhooge@UGent.be)

Functional copolymers show a large potential for a wide range of application fields, including polymer therapeutics, antibody drug delivery, and hydrogels[1–13]. Essential is the use of a low functional comonomer amount (up to 20 mol %) to not only ensure industrial implementation but to also retain the properties of the starting or non-functional homopolymer. For example, the aim can be to preserve the polymer stealth behavior while still enabling polymer modification.

The paramount questions that arise are (i) how the functional groups are spread over the different polymer chains and (ii) how side reactions can disturb the functionalization pattern governed by the main reactions. Quality assessment is currently performed based on simplified reaction schemes and average functionalization degrees so that the distributed nature of the functional monomer incorporation is significantly wiped out. Chain-to-chain deviations can nonetheless be expected based on recent theoretical studies, with a distribution both in chain length (CL) and composition (e.g., the number of functionalities FUNC)[14–19]. It is therefore crucial to develop a tool allowing to quantify the fraction of non-functionalized ($f_{nonfunctionalized}$), single or even multiple functionalized chains for a broad range of functionalization chemistries.

Within the class of chain-growth polymerization, two important techniques for the synthesis of functional polymers are cationic ring opening polymerization (CROP) of 2-oxazolines[20–23] and reversible deactivation radical polymerization (RDRP) of vinyl monomers (e.g., styrene and (meth)acrylates). For CROP, poly(2-alkyl-2-oxazoline)s (PAOx) with methyl ester functionalized side chains are specifically interesting as they can undergo a direct amidation or can be hydrolyzed to a carboxylic acid, making them versatile functional copolymers for conjugation[6,7,24–30]. For RDRP, an important technique is atom transfer radical polymerization (ATRP) with e.g., N-propyl maleimide or glycidyl methacrylate as functional comonomer prone to conjugation or further chemical modification[31,32].

The reaction scheme for the CROP of 2-oxazolines is given in Fig. 1a, including main and side reactions[33]. Being a chain-growth polymerization, CROP conceptually consists of chain initiation ($k_i$), propagation ($k_p$), and termination ($k_t$)[34]. Chain initiation is realized by a nucleophilic attack of the nitrogen of a cyclic imino ether (2-oxazoline) monomer (M) to an electrophilic center of for instance an alkylating reagent (I; initiator) such as methyl tosylate, creating a low molar mass oxazolinium species ($P_1$). This cationic species exhibits a new electrophilic center on the 5-position, enabling the next nucleophilic attack of monomer, inducing the isomerization of the existing oxazolinium from a cyclic imino ether to an amide. This results in the formation of a macro-oxazolinium species ($P_i$; i: chain length), being reactive for further nucleophilic attack and polymerization. The propagation is continued until all monomer is consumed and/or a nucleophilic terminator (e.g., methanolic sodium hydroxide) is introduced to the polymerization mixture (formation of $P_i$' in Fig. 1a). Side reactions (dashed box in Fig. 1a) can also take place, such as chain transfer to monomer ($k_{trM}$) proceeding via $\beta$-elimination. The formed initiator fragment can act as a chain carrier and the formed macromonomer ($D_i$) fragment can compete with conventional monomer for chain growth. Upon macropropagation ($k_{pm}$) a living macrospecies is formed with a cationic centre in the middle of the chain, a so-called mid-chain cationic macrospecies, which upon further monomer addition leads to branch formation.

Figure 1b depicts the core reactions for ATRP synthesis with vinyl monomers, where cations are switched for radicals as chain carriers but similar reactions take place. Chain initiation ($k_i$) is now possible after the reversible activation ($k_{(d)a,i}$) of an alkyl halide initiator ($R_0X$) by a lower oxidation state (copper) catalyst

($M_t^nL_yX$). The formed radical ($R_1$) propagates ($k_p$) leading to the formation of a macroradical $R_i$, which can be deactivated ($k_{da}$) by the higher oxidation state catalyst ($M_t^{n+1}L_yX_2$) into a dormant polymer chain with end-group functionality X ($R_iX$). An important side reaction is termination ($k_t$), leading to the formation of dead polymer chains with no X (dashed box again).

Here we illustrate that the synergetic combination of experimental and in silico characterization provides a unique quality label for CROP of a non-functional 2-oxazoline ($M_1$) and a functional 2-oxazoline with a methyl ester side group ($M_2$). This label, which is based on the construction of a so-called functionality-chain length distribution (FUNC-CLD), allows to accurately calculate the functionalization pattern and is codetermined by side reactions such as $\beta$-elimination (Fig. 1a: dashed box). We show that inherent limitations exist due to the stochastic nature of chain-growth polymerization and that redesigning of the reaction conditions (e.g., the initial functional comonomer amount and the polymerization temperature) and the chemical structure of the comonomers allows the synthesis of highly functional PAOx chains. The reported results and insights are also relevant for the synthesis of functional polymeric materials employing other chemistries, as illustrated with ATRP as RDRP technique.

## Results

**Concept of functionality-chain length distribution.** Kinetic Monte Carlo (kMC) modeling allows to track several molecular characteristics along the synthesis[16,18,19,35–39]. In particular, kMC modeling allows to calculate the number of chains with a certain chain length (CL; equivalently the total number of monomer units) and a given number of functional comonomer units (FUNC value) so that in the context of the present work a bivariate functionality-chain length distribution (FUNC-CLD) can be constructed, with D highlighting the distributed nature of the variates in front[37,38].

Such distribution is shown in Fig. 2 with in the middle the 3D view and at the bottom the 2D projection[36,39], considering—at complete monomer conversion—the theoretical case of a perfect statistical copolymerization with a quasi-instantaneous growth of active chains and no side reactions and equimolar initial amounts of functional and non-functional comonomer, and aiming at a target degree of polymerization (DP) of 100. It follows that the dominant contribution is covered by chains with a length of 100 and a FUNC of 50 but still a Gaussian scatter in both length and FUNC values is obtained, highlighting the unavoidable stochastic nature of chain-growth polymerizations.

The marginal distributions of FUNC-CLD (so integration over FUNC or CL) are by definition the functionality distribution (FUNCD; top right arrow) and the chain length distribution (CLD; top left arrow), as also illustrated in Fig. 2 (equimolar initial conditions; target DP of 100). Distributed properties for the incorporation of the functional comonomer can thus uniquely be obtained, transcending the traditional description of only average characteristics such as the number average chain length ($x_n$) or the average functionalization ($\mu_{FUNCD}$), i.e., the average number of functional monomer units per chain. These averages can still be calculated with the kMC model developed in the present work (labels on x-axes in Fig. 2).

In Supplementary Table S15 the associated results for non-equimolar initial conditions and other target DPs are included. It is demonstrated that even for statistical copolymerizations without side reactions non-functional chains (FUNC = 0) are formed, in particular for too low initial functional comonomer amounts and target DPs. Such deviations are further enlarged if side reactions, as occurring in practice, are also considered as further demonstrated focusing first on CROP and then ATRP synthesis.

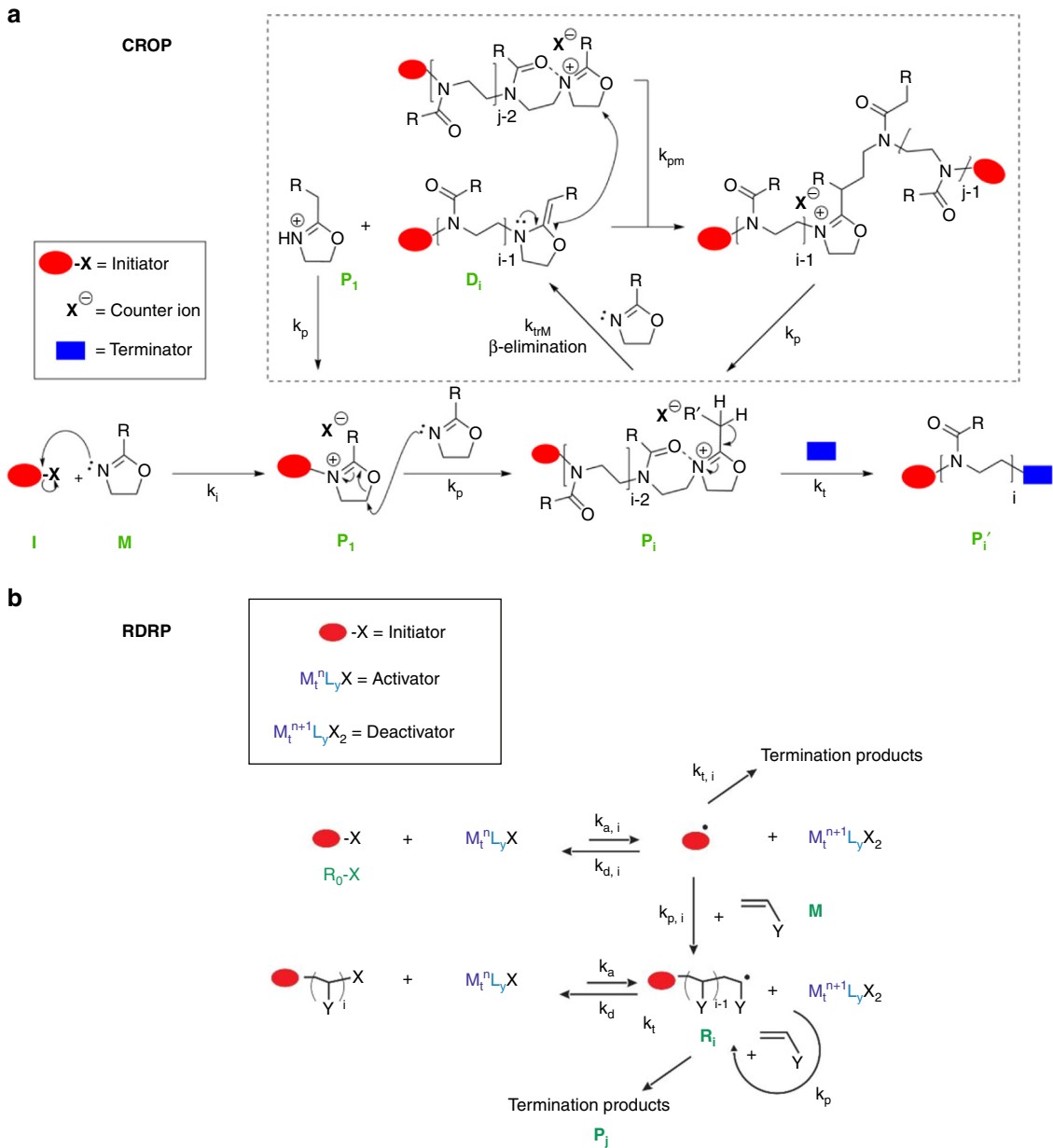

**Fig. 1** Principle of cationic ring opening and reversible deactivation radical polymerization. **a** reactions for cationic ring opening polymerization (CROP) of 2-oxazolines (M) with initiator I; $k_{i,p,t,trM,pm}$: rate coefficient of chain initiation leading to formation of $P_1$, propagation leading to formation of $P_i$ (i: chain length; i > 1), termination leading to formation of $P_i'$, $\beta$-elimination leading to formation of macromonomer $D_i$, and macropropagation leading to formation of mid-chain cationic macrospecies with further propagation leading to branching; red oval: initiator part; $X^-$: counter ion; blue box: terminating agent, e.g., methanolic sodium hydroxide; dashed box: side reactions; R: e.g., methyl. **b** reactions for atom transfer radical polymerization (ATRP), a key reversible deactivation radical polymerization (RDRP) technique, with vinyl monomer (M); $k_{p,i,p,t(,i),a(,i),da(,i)}$: rate coefficient of chain initiation leading to formation of $R_1$, propagation to $R_i$, termination toward $P_i$, activation, and deactivation with (,i) referring to the ATRP initiator $R_0X$; X: end-group functionality; activator/deactivator: $M_t^nL_yX/M_t^{n+1}L_yX_2$; $M_t$: transition metal; L: ligand; dashed box: side reactions

**Relevance functional 2-oxazoline amount on incorporation.** For the CROP of the comonomer pairs MeOx/C2MestOx, EtOx/C2MestOx, MeOx/C3MestOx, and EtOx/C3MestOx, which involves side reactions (Fig. 1a; dashed box), Supplementary Fig. 7-10 show an acceptable agreement of experimental and simulated comonomer conversion, $x_n$, and dispersity copolymerization data under equimolar conditions ($[M_1]_0 = [M_2]_0$), thereby confirming the physical relevance of the kinetic parameters in Supplementary Table S1-4. Equimolar conditions are deliberately considered here to enhance parameter sensitivity toward the copolymerization specific reactions and to allow for—at least—a qualitative in silico

ranking of the functionality qualities later on in which lower functional monomer loadings are covered.

This agreement is also illustrated in Fig. 3 (identical to Supplementary Fig. 7) that shows comparisons between theory and experiment for the comonomer pair MeOx/C2MestOx at four polymerization temperatures: 80 (yellow), 100 (blue), 120 (green), and 140 (purple) °C. The target DP, which is here defined as the initial molar ratio of CROP initiator to monomer, is equal to 100. Additionally, the simulated fractions of branched chains (dashed lines) and of macromonomer chains (full lines), as formed via chain transfer to monomer (Fig. 1a; dashed box), are provided.

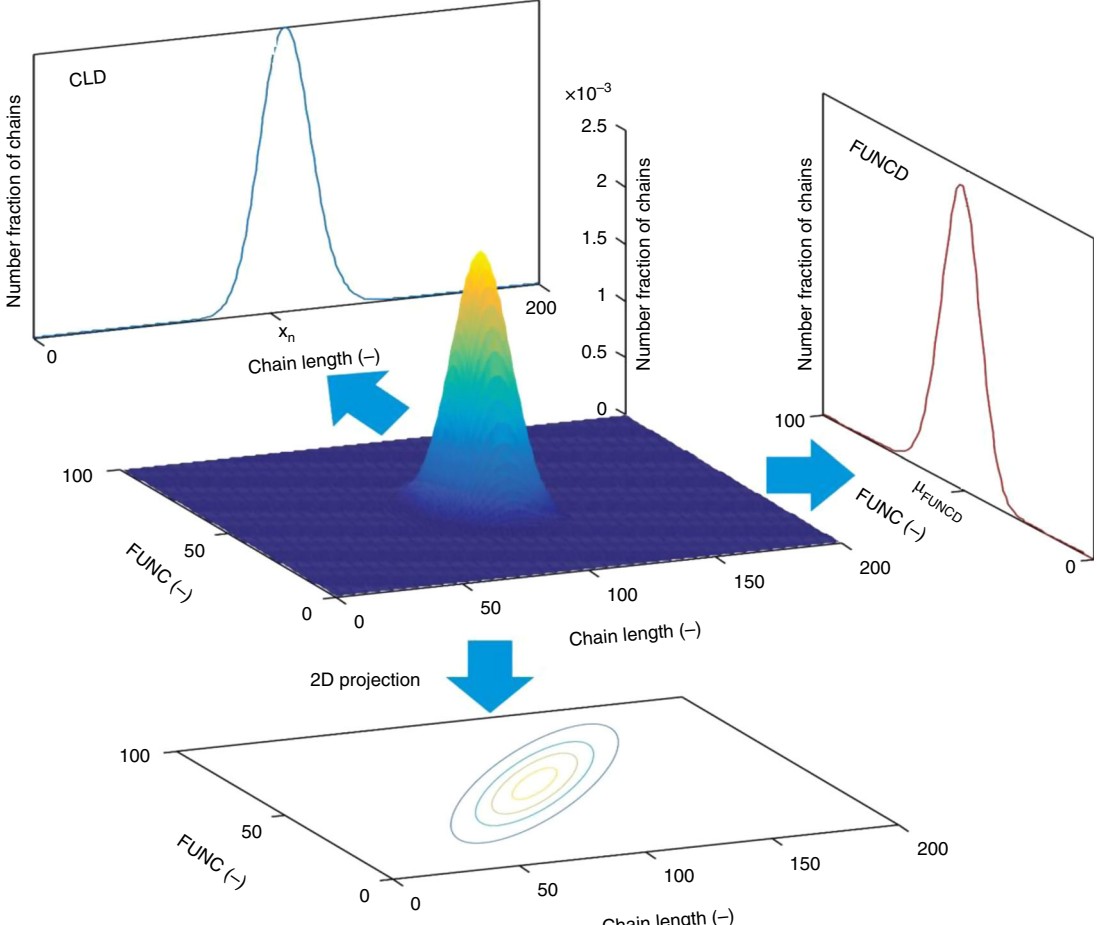

**Fig. 2** Principle of in silico generated functionality-chain length distribution (FUNC-CLD). This is illustrated for a theoretically perfect functionalization chemistry (no side reactions and a fast initiation stage) starting with equimolar amounts of functional and non-functional comonomer and aiming at a target degree of polymerization (DP) of 100; middle: FUNC-CLD in 3D view; bottom FUNC-CLD as a 2D projection. The bivariate FUNC-CLD highlights the fraction of chains with a given chain length (CL) and a given number of functional monomer units (FUNC); for this ideal case most probable are chains with a length of 100 and a FUNC of 50; also shown are the two marginal distributions that can be constructed out of FUNC-CLD by integration over CL or FUNC: (i) the chain length distribution (CLD) and (ii) the functionality distribution (FUNCD). The conventionally used mean/average values of these distributions are also included: for CLD this is the number average chain length ($x_n$) and for FUNCD the average functionalization ($\mu_{FUNCD}$)

It follows from Fig. 3a that a slower incorporation of C2MestOx (dashed lines; closed symbols) is obtained in the copolymerization with MeOx, despite the higher homo-propagation rate coefficient for this comonomer ($1.66 \cdot 10^{-1}$ (C2MestOx) vs. $1.45 \cdot 10^{-1}$ (MeOx) L mol$^{-1}$ s$^{-1}$) as also deducible from the comparison of the MeOx and C2MestOx homopoly-merization data in Supplementary Fig. 1-6 the and literature data[40,41]. Hence, cross-propagation between the MeOx monomer and the C2MestOx macrocation is fast, thereby suppressing chain transfer involving the same species. Figure 3c shows that the dispersity increases already at low monomer conversion and this more clearly at higher temperatures, which can be attributed to the higher relevance of chain transfer (Fig. 3d; full lines). This chain transfer is a gradual process but the formed macromono-mers do not significantly take part in macropropagation as the fraction of branched chains is well below 3 mol % (Fig. 3d; dashed lines). Note that there is some discrepancy between experimental and simulated dispersity data. However, a combined experimental and theoretical study with a sufficiently high overall number of data points allows to strongly reduce experimental uncertainties, taking into account that Arrhenius behavior can be highly expected, which is fully grasped by a kinetic model.

Conventional analysis of the comonomer incorporation in a copolymerization process focuses in addition mostly on the instantaneous average copolymer composition $F_{A,inst}$[42], which reflects the average amount of comonomer A incorporated in all active chains at a certain moment for a given molar fraction of A in the reaction mixture ($f_A$).

For CROP of MeOx and C2MestOx traditional focus is therefore on $F_{MeOx,inst}$ or $F_{C2MestOx,inst}$, i.e., the average amount of MeOx or C2MestOx being instantaneously incorporated. Figure 4a shows the variation of $F_{C2MestOx,inst}$ with varying $f_{C2MestOx}$, with $f_{C2MestOx,0}$ ($f_{C2MestOx}$ at $t = 0$) equal to 10 (purple dots) and 50 (black dots) mol % (140 °C; target DP of 100). The changes for $f_{C2MestOx}$ and $F_{C2MestOx,inst}$ as a function of the (overall) monomer conversion are provided in Fig. 4b, c. Figure 4a shows that for both the 10 and 50 mol % initial functionality loading the well-known terminal Mayo-Lewis[41,42] behavior is followed:

$$F_{C2MestOx,\,inst} = \frac{r_1 f_{C2MestOx}^2 + f_{C2MestOx}(1 - f_{C2MestOx})}{r_1 f_{C2MestOx}^2 + 2f_{C2MestOx}(1 - f_{C2MestOx}) + r_2(1 - f_{C2MestOx})^2}$$

(1)

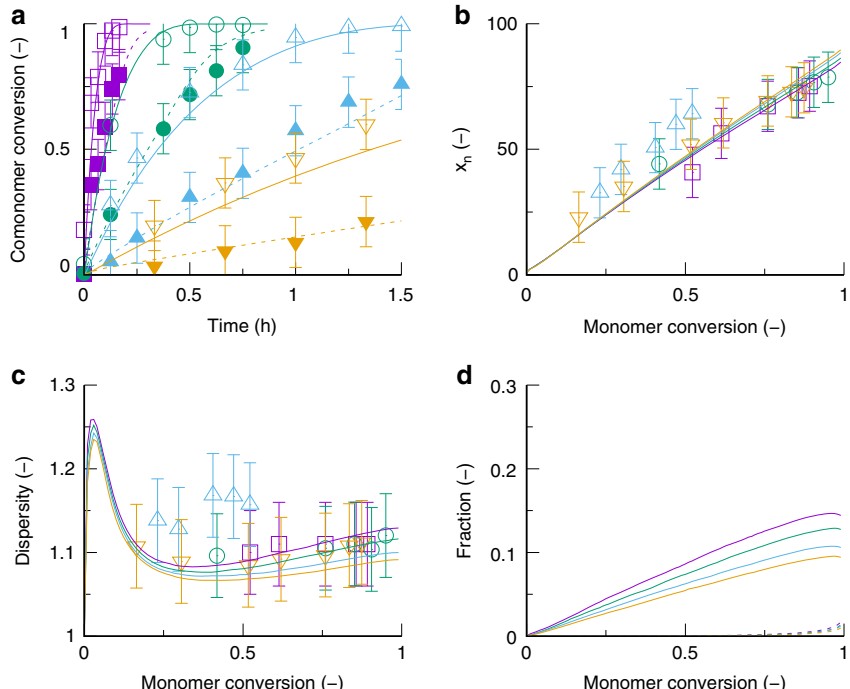

**Fig. 3** Model validation for CROP of MeOx and C2MestOx under equimolar conditions. **a** Comonomer conversion profiles (closed/open symbols: functional/non-functional monomer; simulations: full/dashed lines). **b** number average chain length ($x_n$). **c** dispersity. **d** macromonomer fraction (full lines; top) and fraction of branched chains (dashed line; bottom, very close to x-axis) as a function of (overall) monomer conversion; total monomer concentration: 3 mol $L^{-1}$; solvent acetonitrile; target DP = 100; 80 (yellow), 100 (blue), 120 (green), and 140 °C (purple)); lines: simulations; symbols: experimental data[43]. Model validation for the other three comonomer pairs in Supplementary Fig. 7-10; the reported error bars relate to the standard deviations following from repeat experiments

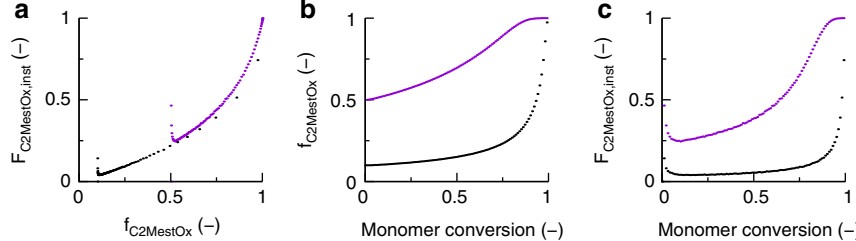

**Fig. 4** Conventional analysis of 2-oxazoline comonomer incorporation. **a** Average instantaneous functionalization ($F_{C2MestOx,inst}$) as a function of the feed composition based on the functional comonomer ($f_{C2MestOx}$). **b** $f_{C2MestOx}$ as a function a function of the (overall) monomer conversion. **c** $F_{C2MestOx,inst}$ as a function the (overall) monomer conversion; CROP of MeOx and C2MestOx under equimolar initial conditions (conditions: Fig. 3); purple and black dots: initial amount of $C_{2MestOx}$ ($f_{C2MestOx,0}$) of 50 and 10 mol %; for **a** almost coinciding with terminal Mayo-Lewis equation (Equation (1)); cumulative average functionalization ($F_{C2MestOx}$) evolutions in Supplementary Fig. 13; more compositional drifting for a lower initial C2MestOx loading. However, the chain length history is ignored so that a biased description is obtained, justifying the need for the more detailed method as introduced in Fig. 2 and applied in what follows

Figure 4b shows that the monomer feed is enriching in C2MestOx ($f_{C2MestOx} > f_{C2MestOx0}$) as the polymerization progresses and this more gradually for the 50 mol % case (purple dots). In agreement with this observation, Fig. 4c shows for the 50 mol % case a more gradual shift from lower to higher $F_{C2MestOx,inst}$ with increasing monomer conversion. For the 10 mol % case (black dots in Fig. 4c), $F_{C2MestOx,inst}$ is approximately constant for the first half of the polymerization—as defined on a monomer conversion basis—at a value lower than the targeted 10 mol %. Only at the final stage, the copolymer chains are being enriched in C2MestOx, but unfortunately much more than the targeted 10 mol %. For all monomer conversions there is thus a strong mismatch with the targeted value and a gradient-like incorporation of the functional comonomer takes place as expected based on the reactivity ratios (Supplementary Table 6 and 7[43–46]).

Hence, Fig. 4 reveals that a stronger compositional drift results for a lower $f_{C2MestOx0}$, as supported by the cumulative average functionalization ($F_{C2MestOx}$) evolutions covered in Supplementary Figs. 13 and 14. Supplementary Fig. 13 further highlights that a cessation of the polymerization at lower monomer conversion is not recommended as mismatches at lower monomer conversion can only be compensated for at higher monomer conversion due to the specific shape of the Mayo-Lewis plot (Fig. 4a). $F_{C2MestOx}$ does, however, not account for the chain length history (e.g., more chains due to chain transfer) as the original active chains are assumed to be continuously growing. The latter is illustrated in Supplementary Fig. 14 with identical $F_{C2MestOx}$ evolutions for a model with and without chain transfer reactions, highlighting that conventional analysis does not provide information on the functionalization per chain (length). As CROP with C2MestOx

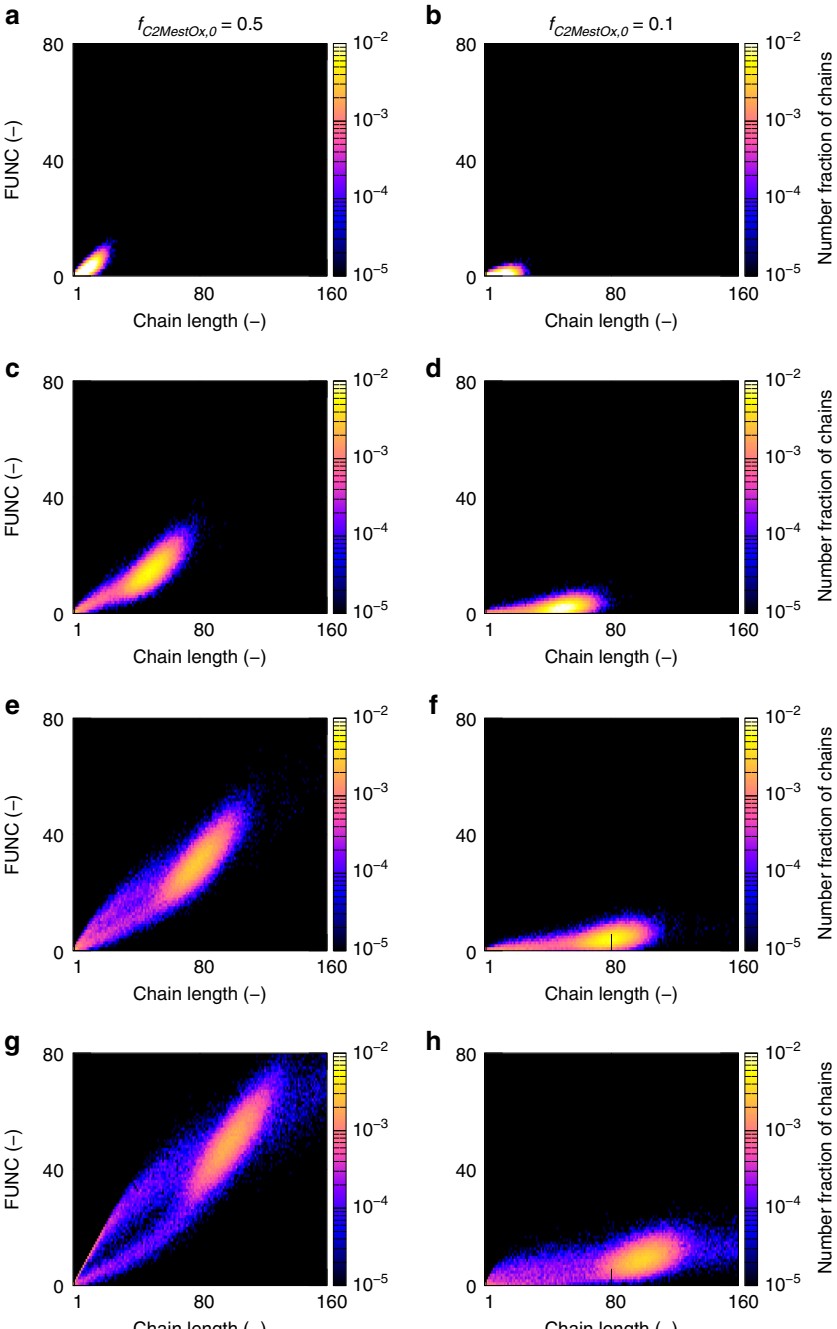

**Fig. 5** Functionalization—chain length distribution (FUNC-CLD) or fingerprint for CROP of MeOx and C2MestOx. Variations are included for an initial amount of C2MestOx ($f_{C2MestOx,0}$) of 0.5 and 0.1 (total monomer concentration: 3 mol L$^{-1}$; solvent acetonitrile; target DP of 100; 140 °C) at (overall) monomer conversion of **a**, **b** 10%, **c**, **d** 50%, **e**, **f** 80%, and **g**, **h** 100%; at final conversion a trimodal fingerprint results

can be subject to chain transfer (Fig. 3d) as well as composition drift (Fig. 4), it is therefore necessary to investigate the distribution of the functional C2MestOx units over the actual chain lengths (Fig. 2: FUNC-CLD).

For the equimolar MeOx/C2MestOx system (140 °C; target DP of 100), the first column of Fig. 5 shows FUNC-CLD at a monomer conversion equal to 10, 50, 80, and 100%. Note that FUNC has a minimum value of 0 as chains can also be non-functional, i.e., poly(MeOx) homopolymer chains. With increasing monomer conversion, the FUNC-CLDs in Fig. 5 (left column) alter from unimodal to multimodal. At a monomer conversion of 10% (Fig. 5a), unimodality results as can be easily identified by the peak in the left bottom corner with a steep transition from higher to lower number fractions. At a monomer conversion of 50% (Fig. 5c), the polymer chains have sufficiently grown and more functional comonomer units have been incorporated, as reflected by a positive sloping of the aforementioned peak. In addition, a second peak emerges at the low chain lengths and C2MestOx amounts, yielding a bimodal FUNC-CLD. At a monomer conversion of 80% (Fig. 4e), a third C2MestOx-rich peak develops in the region of low chain lengths but higher FUNC values, resulting in a trimodal FUNC-CLD, which can be seen as a fingerprint for the final copolymer product at a monomer conversion of 100% (Fig. 5g).

This trimodal fingerprint can be explained considering the CROP mechanism (Fig. 1a), as illustrated in the top row of Fig. 6.

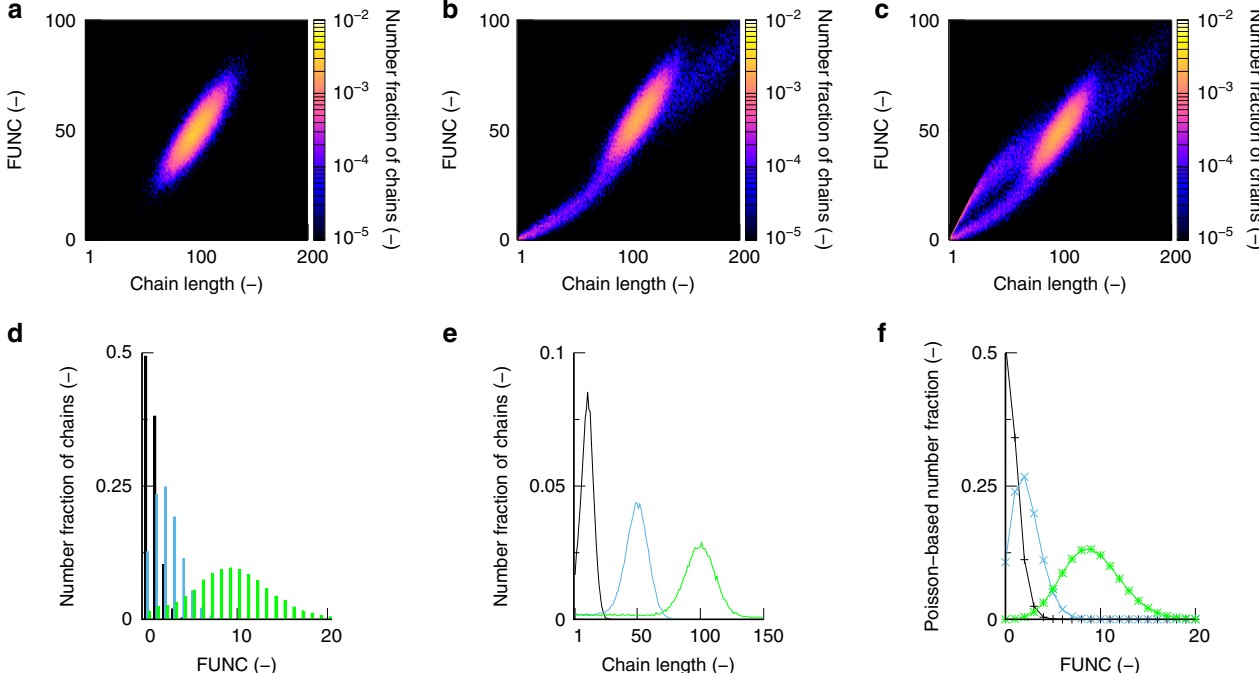

**Fig. 6** Understanding the distributed nature of functionalization with 2-oxazoline. Top row: impact of side reactions on FUNC-CLD under equimolar conditions (total monomer concentration: 3 mol L$^{-1}$; solvent acetonitrile; target DP of 100; 140 °C; overall monomer conversion of 100%) **a** formally no chain transfer. **b** with chain transfer but formally without chain initiation of the formed oxazolinium fragment. **c** with chain transfer and the latter chain initiation (identical to Fig. 5g) Bottom row marginal distributions of FUNC-CLD depicted in Fig. 5h (10 mol % initially for C2MestOx) **d** functionalization distribution (FUNCD) and **e** chain length distribution (CLD) at an overall conversion of 10% (black), 50% (blue), and 100% (light green); additionally **f** Poisson distributions based on mean values of the FUNCDs ($\mu_{FUNCD}$) in **d**; the latter distributions are approximate and a complete FUNCD as extractable from FUNC-CLD is the preferred route

In case chain transfer is formally removed from the kinetic model, a very clean monomodal fingerprint results with the majority of the chains possessing *ca.* 50 alkyl substituted and methyl ester substituted 2-oxazoline units in agreement with the initial feed composition (Fig. 6a). Hence, here the functionalization is behaving as in the theoretical perfect case of Fig. 2. Upon the consideration of chain transfer but not the chain initiation by the formed oxazolinium fragment as a result of the former reaction, the peak at the low chain length and C2MestOx amount becomes visible (Fig. 6b). If the chain initiation after chain transfer is also accounted for, the missing top arm of the actual fingerprint is generated, i.e., Figs. 6c, 5g become identical.

Hence, the chain transfer process not only increases the dispersity (broadening parallel to the *x*-axis in Fig. 5g), but also leads to a copolymer product with a stronger compositional drift per chain length (broadening parallel to the *y*-axis in Fig. 5g). It should be stressed that this insight with strong deviations from the ideal theoretical case (Fig. 2) can only be covered with the bivariate modeling (FUNC-CLD) strategy of the present work.

A comparison of the left and right column of Fig. 5 allows to deduce how the initial C2MestOx amount influences the FUNC-CLD (50 vs. 10%). For both loadings, at a monomer conversion of 100%, the major peak is located at a chain length of 100, i.e., the target DP, with an average functional comonomer incorporation equal to the initial C2MestOx feed. For a decreasing initial amount, less chains with a high C2MestOx incorporation are formed, as can be expected. In addition, a kind of folding can be observed as the two-arm structure at the low chain length and C2MestOx region becomes less visible. A closer inspection reveals that at a lower initial loading a much flatter and stretched-out FUNC-CLD is obtained with in particular a stronger spread toward the *x*-axis (see also Supplementary Fig. 15; bottom row,

also case of 5%). For a lower initial functional monomer loading, this indicates a significant contribution of the non-functionalized chains and a higher compositional drift per chain, which is in agreement with the discussion of the $F_{C2MestOx, inst}$ (refer to Fig. 4), and the $F_{C2MestOx}$, evolution (Supplementary Figs. 14 and 15).

With the bivariate FUNC-CLD available any derived univariate or average characteristic is within reach. For example, the functionality distribution (FUNCD; cf. right top arrow in Fig. 2) can be retrieved. This distribution reflects the (number) fraction of chains possessing a given number of functional comonomer units (0, 1, 2, …). For MeOx-C2MestOx (140 °C; target DP of 100; 10 mol % C2MestOx initially; FUNC-CLD: Fig. 5h), Fig. 6d shows FUNCD at distinct monomer conversion (10, 50, and 100%), with the corresponding CLDs provided in Fig. 6e. With increasing monomer conversion both FUNCD and CLD shift to the right but they have a different shape, indicating that a conventional SEC trace cannot be used to validate the success of the copolymerization with respect to its functionalization.

An approximation of FUNCD by a Poisson distribution (Fig. 6f) with the Poisson parameter being the mean value of FUNCD ($\mu_{FUNCD}$; the average number of functional monomer units per chain) is also not afforded. The Poisson distribution is similar as the simulated FUNCD at lower monomer conversion (black line in Fig. 6f vs. black bars in Fig. 6d) but narrower than the FUNCD at higher monomer conversion (green line in Fig. 6f vs. green bars Fig. 6d). The Poisson distribution neglects side reactions such as chain transfer. Hence, care must be taken upon using analytical expressions to assess the success of a functional copolymerization process involving side reactions. It is thus again shown that a complete characterization by FUNCD as extractable from FUNC-CLD is the preferred route.

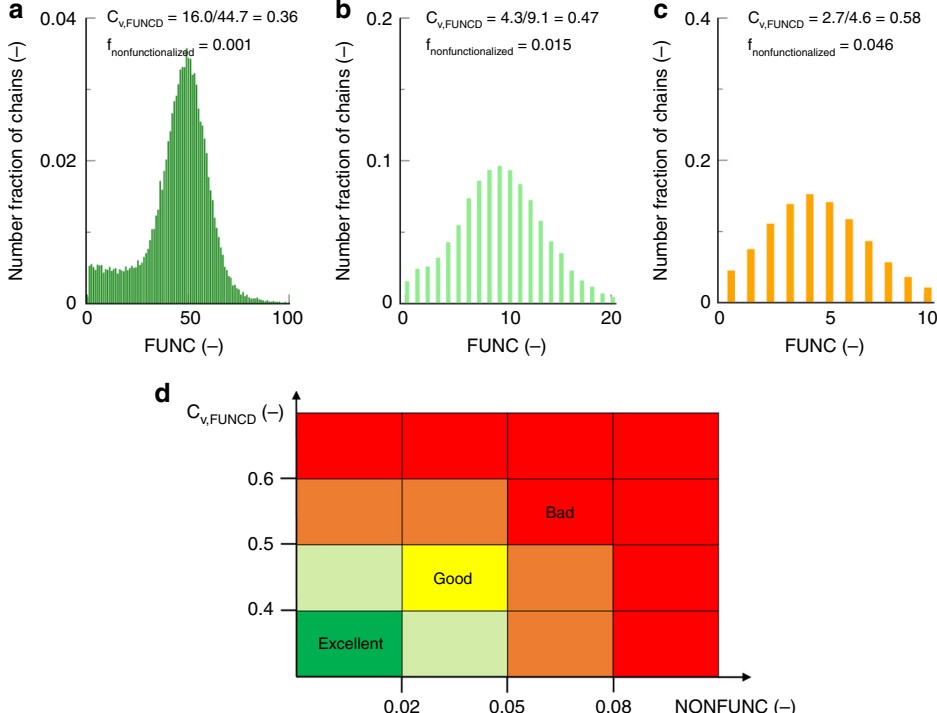

**Fig. 7** Translation of functionalization distribution in two derived properties. **a–c** Functionality distribution (FUNCD) from FUNC-CLDs in Supplementary Fig. 15 ($f_{C2MestOx,0}$ = 50/10/5 mol %; 140 °C; target DP of 100). **d** associated color coding for a fast (guide of the eye) comparison of a functionalization success: full details see Supplementary Discussion (bad to excellent functionalization: dark red to dark green, with intermediate colors orange, yellow and light green); only with the extremely high initial amount of 50% an excellent functionalization (dark green) is obtained

The compositional drift observed in the FUNCD can be further quantified by calculating the coefficient of variation ($C_v$), which is defined as:

$$C_{v,FUNCD} = \frac{\sigma_{FUNCD}}{\mu_{FUNCD}} \qquad (2)$$

in which $\sigma_{FUNCD}$ is the standard deviation for FUNCD. Together with the fraction of non-functionalized chains, which is denoted as $f_{nonfunctionalized}$ (fraction of chains with FUNC = 0), the $C_{v,FUNCD}$ value allows to map the functionalization quality in a straightforward manner.

To enable a fast detection of the relation between reaction conditions and the functionalization quality, the FUNCDs, as obtained upon complete monomer consumption, are colored. The color code of Fig. 7d is followed with a formal gradient from bad (dark red color) to excellent (dark green color). Intermediate colors are light green, yellow, and orange. Note that the $C_{v,FUNCD}$ and $f_{nonfunctionalized}$ boundaries defining these color changes are arbitrary but at least allow a ranking of functional copolymers impossible based on experimental research only. The key values (transition to green color; 0.5 for $C_{v,FUNCD}$ and 0.05 for $f_{nonfunctionalized}$) are also common values used in the general field of statistics. Moreover, the color trends are relevant for application as purification by preparative SEC is impossible and preparative liquid chromatography for polymers is cumbersome.

The effect of the initial C2MestOx amount on FUNCD is illustrated in Fig. 7 (monomer conversion of 100%; target DP of 100; 140 °C), with an excellent functionalization (dark green FUNCD) only for the case with an extremely high initial amount of functional comonomer (50 mol %). In agreement with the trend for FUNC-CLD (Fig. 5), a lower initial amount leads to a higher $f_{nonfunctionalized}$. Upon lowering the initial C2MestOx amount (left to right in Fig. 7), $C_{v,FUNCD}$ and $f_{nonfunctionalized}$ increase with a factor of *ca.* 1.5 and 50, highlighting a reduction of

the functionality incorporation. For the lowest loading of 5 mol % (Fig. 7c), the NON-FUNC fraction is even very close to the limiting value of 0.05 to obtain an undesired labeling, the same being true for $C_{v,FUNCD}$ with a limiting value for bad labeling of 0.5, explaining the non-green color for this FUNCD. Hence, the color shift in Fig. 7 (dark green to light green to orange) indicates the need for a threshold value for the initial C2MestOx monomer loading, at least at a target DP of 100 and 140 °C.

Supplementary Fig. 16 (10 mol % case) shows that $C_{v,FUNCD}$ (Equation (2)) is rather constant as a function of monomer conversion as similar increases for $\sigma_{FUNCD}$ and $\mu_{FUNCD}$ are obtained. A comparison of $\mu_{FUNCD}$ values is thus not recommended to evaluate the success of functionalization as it does not take into account the variations of the FUNCD shape. The latter is consistent with the incapability of the Poisson distribution to properly reflect the functionalization process (refer to Fig. 6f). Supplementary Fig. 16 also allows to deduce that $f_{nonfunctionalized}$ reaches low values only at high monomer conversion, in agreement with the aforementioned steep increases of $F_{C2MestOx}$ at the end of the copolymerization process (refer to Fig. 4).

**Design for narrow functionality distributions.** Improved copolymer product quality can be targeted by the experimentalist through varying the chemical nature of the reactants and/or the initial concentrations (target DP and initial loading of functional comonomer) and temperature. This can be a time consuming procedure and as shown above the full evaluation of functional monomer incorporation is not straightforward with common analytical tools. Model-based design can therefore be very supportive and efficient for designing improved experimental protocols[36,47–49]. Moreover, as highlighted above, only FUNC-CLDs as accessible through advanced kMC simulations allow to quantify the functionalization quality (Figs. 5–7) while conventional analysis is limited to average characteristics (Fig. 4).

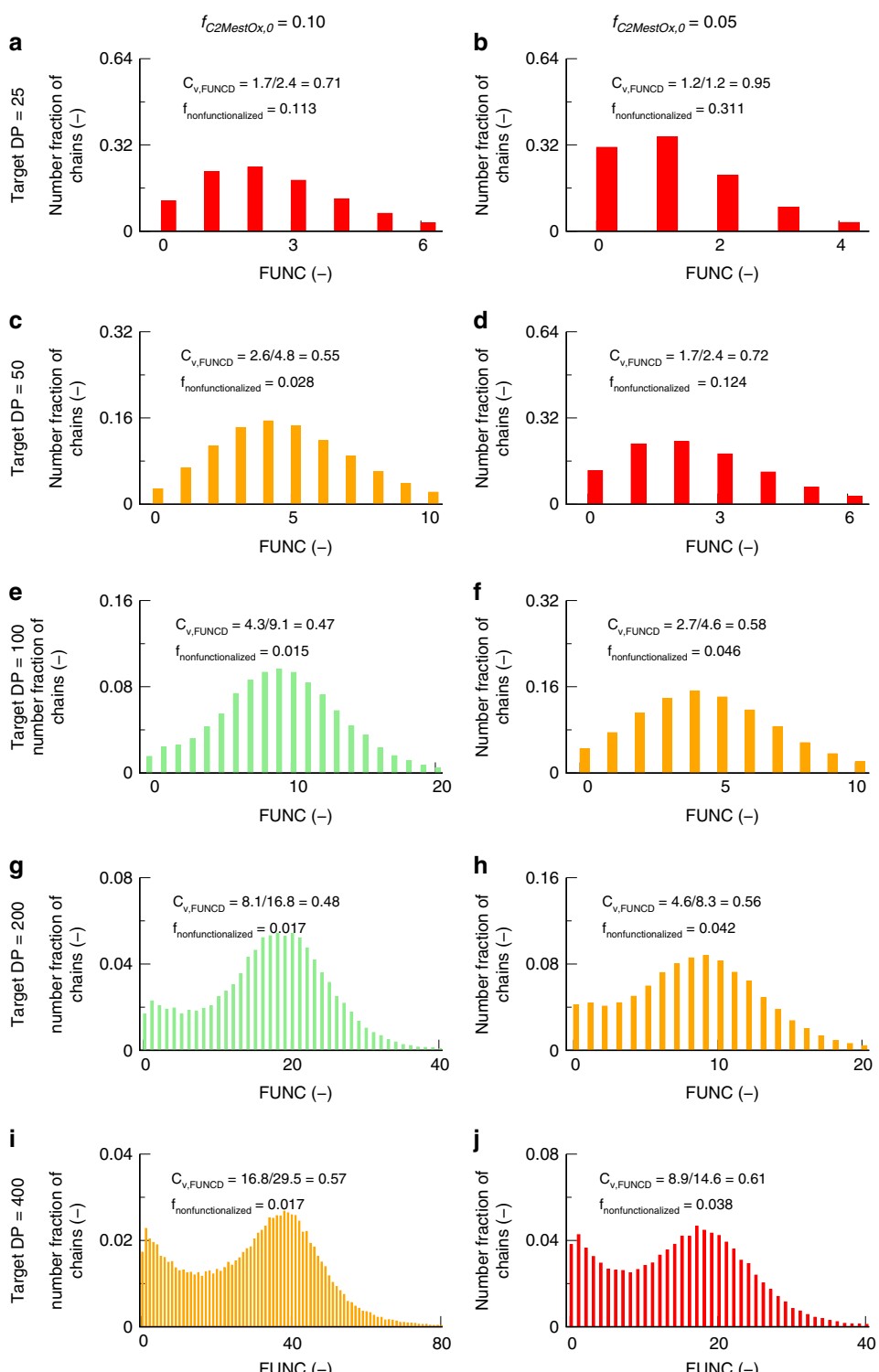

**Fig. 8** Relation reaction conditions and functionality control. Effect of initial functional comonomer amount (columns; 10 (**a**, **c**, **e**, **g**, **i**) and 5 % (**b**, **d**, **f**, **h**, **j**)) and target DP (rows; 25 (**a**, **b**), 50 (**c**, **d**), 100 (**e**, **f**), 200 (**g**, **h**), and 400 (**i**, **j**)) on the functionality distribution (FUNCD) for CROP of MeOx and C2MestOx (total monomer concentration: 3 mol L$^{-1}$; solvent acetonitrile; Temperature = 140 °C; overall monomer conversion: 100%); color coding: according to Fig. 7d as visual aid; FUNCDs constructed out of FUNC-CLDs in Supplementary Fig. 17; only intermediate target DPs allow a reasonable functionalization (light green or yellow color)

Focusing on the comonomer pair MeOx/C2MestOx and a reaction temperature of 140 °C, Fig. 8 shows for 2 initial functional comonomer amounts (5 and 10 mol %; left and right column), FUNCD at a monomer conversion of 100% for a target DP of 50, 100, 200, and 400 (top to bottom). These FUNCDs were constructed out of the fingerprints, i.e., FUNC-CLDs, given in

Fig. S17 of the Supplementary Information. The best functionalization is obtained for intermediate target DPs (100–200), as low target DPs result in too high $f_{nonfunctionalized}$ and too high $C_{v, FUNCD}$ values, and too high target DPs result in too high $C_{v,FUNCD}$ values, due to chain transfer reactions, despite the excellent $f_{nonfunctionalized}$ values.

Figure 8 also shows that for a lower initial functional comonomer amount (right column), a broader FUNCD (higher $C_{v,FUNCD}$) results at every target DP (given row), in agreement with the results in Fig. 5 (target DP of 100). No dark green FUNCDs can although be detected in Fig. 8, highlighting that—for the selected reactions conditions and comonomer pair—the functionalization quality is at most very good but not excellent (refer to color coding in Fig. 7). It can be seen that only for the higher initial C2MestOx amount ($f_{C2MestOx,0} = 10$ mol %; left column) a reasonably defined FUNCD (light green color) is obtained upon the consideration of a sufficiently low target DP (target DP of 100 or 200). For the lower initial C2MestOx amount (5 mol %; right column), at these intermediate optimal target DPs, a moderate control over FUNCD is only obtained (orange FUNCDs).

For a given initial functional comonomer amount (a given column in Fig. 9), $f_{nonfunctionalized}$ lowers toward a plateau value for higher target DPs (0.017 (left column; 5 mol % case) and 0.04 (right column; 10 mol % case). The latter is a direct consequence of the different competition between propagation and chain transfer for each macrocation. Propagation is intrinsically faster and, hence, chain transfer only occurs after a certain number of comonomer incorporations. As under typical conditions C2MestOx is the limiting comonomer in the initial feed a certain amount of short non-functionalized or MeOx homopolymer chains are always formed. Indeed, Supplementary Figs. 18 and 19 indicate that upon doubling the initial C2MestOx amount, so making this monomer somewhat less a limiting reactant, the maximum chain length for the non-functionalized chains becomes ca. twice as low. For too low target DPs, however, the number of monomer units in a chain is too short as such, explaining the significant deviation from the high target DP plateau value (higher $f_{nonfunctionalized}$ values) in Fig. 8.

The reason why in Fig. 8 intermediate target DPs only result in low $C_{v,FUNCD}$ values can be understood by noting two opposite effects. As explained in our previous work[50] on CROP of MeOx and PhOx, the importance of chain transfer decreases for lower target DPs. Such DPs result in lower lifetimes for the macrocations and therefore they are less likely to undergo chain transfer to monomer (Supplementary Fig. 19). However, the calculation of an average characteristic becomes less meaningful for lower target DPs[15,49]. This can be understood by examining the extreme case of very low target DPs with very steep variations in the tail of FUNCD.

The simulated data in Fig. 8 also show that for experimental validation one should focus on low target DPs and low initial loadings of functional monomer, as the associated $f_{nonfunctionalized}$ value is rather high facilitating experimental determination of the fraction of non-functionalized polymer chains. For experimental validation, the side-chain methyl ester groups were hydrolyzed to carboxylic acids groups that were used to immobilize the functional polymer on a solid Wang resin support. The non-functionalized part was washed from the resin and the functional polymer could be recovered from the resin by acid-catalyzed hydrolysis (Supplementary Fig. 27). Using $^1$H NMR spectroscopy the ratio of the side-chain signal intensities before and after the removal of the non-functionalized polymer led to the determination of a value of 28% for the non-functionalized chains for the MeOx-C2MestOx copolymer prepared with a target DP of 25 and an initial loading of 5%. The simulated value 31% in Fig. 8b is close to this experimental value, further highlighting the consistency of the developed approach.

Since chain transfer is more activated than propagation (Supplementary Table 1-4), it could be expected that the relative importance of both reactions can be altered by varying this temperature. Supplementary Table 10 (still comonomer pair MeOx/C2MestOx and the same two initial C2MestOx loadings) shows that lowering the temperature from 140 to 100 °C has an impact on the functionalization quality, in case the optimal target DP of 100 as determined above is used (FUNC-CLDs: Supplementary Fig. 21). The impact is although limited for the selected reaction system.

Hence, it can be postulated that a tuning of the target DP is more beneficial than a tuning of the polymerization temperature, taking into account that a lowering of the latter also results in extended reaction times. For example, to reach a complete monomer conversion—at a target DP of 100—the reaction time goes already up from 0.5 to ca. 4 hours by decreasing the temperature from 140 to 100 °C.

Model-based design is also possible by the variation of the comonomer chemical structures. Supplementary Figure 23 displays FUNCDs at a monomer conversion of 100% for the comonomer pairs MeOx/C2MestOx (top left), EtOx/C2MestOx (top right), MeOx/C3MestOx (bottom left), and EtOx/C3MestOx (bottom right), for a target DP of 100, an initial methyl ester amount of 10 mol %, and at a polymerization temperature of 140 °C. These FUNCDs are obtained from the FUNC-CLDs shown in Supplementary Fig. 22.

The highest functionalization quality is obtained for the MeOx/C2MestOx comonomer pair. Only for that combination a light green color is obtained for FUNCD (very good quality; cf. Supplementary Fig. 23), whereas for the other comonomer pairs a yellow/orange color (good to less good quality) results.

**General guidelines and applicability for other chemistries.** Figure 9 highlights the strength of the simultaneous model-based design of several batch reaction parameters. Focus is on the variation of the target DP and the initial functional monomer loading (refer to Fig. 8), the polymerization temperature (refer to Supplementary Table 10), and the chemical structure of the comonomer pair (refer to Supplementary Fig. 23). The target DP is varied between 100 and 400, bearing in mind that too low

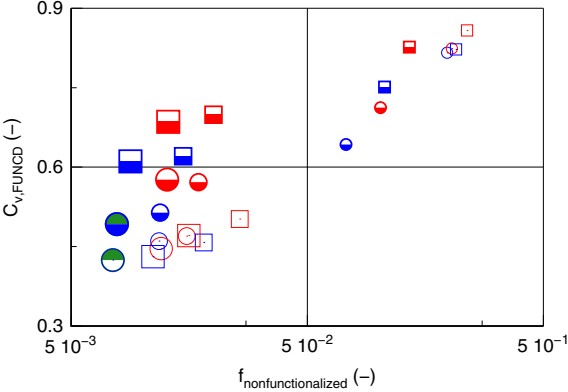

**Fig. 9** Summary of the effect of the batch reaction parameters on the fraction of non-functionalized chains ($f_{nonfunctionalized}$) and the coefficient of variation ($C_{v,FUNCD}$; Equation (2)). This is illustrated for CROP of a functional and a non-functional monomer (total monomer concentration: 3 mol L$^{-1}$; solvent acetonitrile; overall monomer conversion = 100%; FUNCDs constructed out of FUNC-CLDs from of the Supplementary Fig. 24-26 (also table with color coding cf. Figure 7d); in this figure larger symbol: higher initial functional loading (from 2 to 10 to 13 mol %); higher temperature (from 100 to 140 °C) by changing from blue to red color; from open symbol with border with red/blue color to half-filled symbol with the same color to highlight change from target DP of 100 to 400; circle to square: change from MeOx/C2MestOx to EtOx/C2MestOx; model-based design allows to determine optimal conditions for a given target DP (here thus 100 and 400): highlighted as half green symbols

target DPs lead to too high $f_{nonfunctionalized}$ values and too high target DPs result in too high macromonomer formation (too high $C_v$ values), as explained above. The initial functional monomer loading is varied between 2 and 13 mol %. For the sake of comparison also the optimal value from Fig. 8 (10 mol %; 140 °C; light green FUNCDs) is included. The polymerization temperature is varied between 140 and 100 °C, as lower temperatures lead to rather slow polymerizations. C2MestOx is only considered as functional comonomer based on the interpretation of Supplementary Fig. 23. The associated FUNCD-CLDs are given in Supplementary Figs. 24-26, with the $C_v$ and NON-FUNC values also mentioned in Supplementary Table 16.

Semi-batch procedures, as applied for conventional chain-growth copolymerization, are suited if the goal is living functionality control toward complete monomer conversion. Such procedures are relevant if off-spec chains upon a stoppage of the feed stream at high polymerization times are allowed and a functional comonomer can be sufficiently incorporated for a reasonable feed contribution of that comonomer. As shown in Fig. 4a, the Mayo-Lewis profile possesses a less suited shape with high initial functional loadings needed to enable efficient functional comonomer incorporation. Moreover—even under semi-batch conditions—chain transfer will remain a contributor in the disturbance of the polymerization kinetics (cf. Supplementary Fig. 14-15). Under starved feed conditions, semi-batch living/controlled polymerization procedures can be used to obtain the desired structures but at the cost of significantly slowing down the polymerization process.

Upon inspection of Fig. 9 it follows that an initial low loading of 2 mol % C2MestOx results in bad functionalization whatever the other design variables are. At 140 °C and for a loading of 10 mol %—in agreement with the discussion of Fig. 8—the optimal target DP is 100 and—in agreement with the interpretation of Supplementary Fig. 23 the optimal comonomer pair is MeOx/C2MestOx. Lowering the temperature to 100 °C allows to obtain for both comonomer pairs a very good functionalization. For a target DP of 400, a very good functionalization is also obtained for MeOx/C2MestOx in case the temperature is lowered to 100 °C, further confirming the better performance of MeOx/ C2MestOx. Moreover, at a target DP of 100, an excellent functionalization behavior is even obtained for both comonomer pairs if the initial loading is increased from 10 to 13% at 100 °C. For a target DP of 400 (100 °C), the functionalization can even be maintained at very good if the C2/MestOx pair is considered.

Hence, for a given target DP, the PAOx functionalization quality can be properly tuned (sufficiently low $f_{nonfunctionalized}$ and $C_{v,FUNCD}$) by lowering the temperature and increasing the initial functional monomer amount, preferably with the MeOx/C2MestOx comonomer pair. For Fig. 9—with focus on a target DP of 100 and 400—this relates to the green half-filled symbols with the corresponding thus designed FUNC-CLDs provided in Fig. 10a, b. These bivariate distributions strongly deviate from the theoretical one in Fig. 2, further highlighting the relevance of the present work. Once these designed settings are determined further in silico characterization can be applied on the molecular level, for instance the evaluation of the gradient character by the plotting of the individual monomer sequences as covered in previous work[50].

As illustrated in Fig. 10c, d model-based design is also relevant for other (e.g., radical) statistical copolymerizations to prepare functional copolymers. Here focus is on ATRP of styrene with small amounts of N-propyl maleimide (Fig. 10c; addition of 1 equivalent with respect to the initial ATRP initiator molar amount at a styrene conversion of 0.35) and ATRP of 2-ethylhexyl acrylate and glycidyl methacrylate (Fig. 10d; 10 mol % of functional comonomer initially), with the details on the model parameters and validation provided in the Supplementary Discussion (target DP of 100). These systems were chosen to further illustrate the broader value of our modeling approach, as a large number of experimental kinetic parameters were available in literature (see Supplementary Discussion). For the former ATRP case, the monomer chemical structures reflect an alternating-like comonomer incorporation, whereas in the latter case cross-propagation toward the functional comonomer unit is chemically preferred due to the higher stability of tertiary terminal methacrylate radicals compared to secondary terminal acrylate radicals. In both cases, chain-to-chain deviations are still identified that can be attributed to the inherent stochastic nature of the ATRP process (cf. the ideal case in Fig. 2) and additionally non-instantaneous ATRP initiation and unavoidable termination during the early stages of the polymerization. Hence, again the need for a FUNC-CLD based quantification tool acknowledging differences due to side reactions is highlighted.

## Discussion

Unique fingerprints are generated for copolymerizations with a non-functional and a functional comonomer, aiming at the well-defined incorporation of the latter in the individual chains. A fingerprint is a functionalization-chain length distribution (FUNC-CLD) as accessible through $k$MC simulations and allows the a posteriori calculation of $f_{nonfunctionalized}$ and the diversity in the functionalization degree, i.e., the relevance of highly vs lowly functionalized chains ($C_v$ value). Model validation can be performed based on homopolymerization data and copolymerization data under equimolar conditions. These FUNC-CLDs are always characterized by the inherent stochastic behavior of polymerizations and further shaped by side reactions.

For PAOx synthesis with CROP aiming at retaining of the properties of the non-functional homopolymer analogue, it is illustrated that a variation of the chemical structure of the comonomers and/or the reaction conditions allows to tune the functionalization quality. For both low (100) and high (400) target DPs an excellent functionalization quality is obtained with the comonomer pair MeOx/C2MestOx, a polymerization temperature of 100 °C and an initial functionality loading of 13%.

These optimal conditions cannot be identified based on conventional analysis of copolymerization data, as these are limited to average characteristics lacking the complete information on the chain-growth history. In silico generated FUNC-CLD data, on the other hand, allow to fulfill this identification task.

The combined experimental and modeling strategy can be extended to other chemistries and is a crucial step toward the design of functional materials with tailored properties for a broad application field.

## Methods

**Experimental framework.** Methyl ester functionalized copoly(2-oxazoline)s can be prepared by CROP of a (hydrophilic) 2-alkyl-2-oxazoline monomer with a methyl ester functional comonomer, e.g. methyl-3-(2-oxazoline-2-yl) propanoate (C2MestOx; also referred to as 2 methoxycarbonyl-ethyl-2-oxazoline) and methyl-4-(2-oxazoline-2-yl)butanoate (C3MestOx; also referred to as 2-methylcarbonyl-propyl-2-oxazoline). Focus is here on CROP with methyl tosylate (I) as initiator and the comonomer pairs 2-methyl-2-oxazoline (MeOx)/C2MestOx, 2-ethyl-2-oxaoline (EtOx)/C2MestOx, MeOx/C3MestOx, and EtOx/C3MestOx ($M_1$/$M_2$).

Experimental CROP data on comonomer conversion, number average chain length ($x_n$), which is defined as the average total number of comonomer units, and dispersity, at different polymerization temperatures have been taken from experimental studies performed by the Hoogenboom research group[43,51,52]. The reported error bars relate to the standard deviations following from repeat experiments. Based on literature data[50], the measured $x_n$ data have been rescaled with a constant factor (Supplementary Table 9) to compensate for the difference in hydrodynamic volume of the functional (co)poly(2-oxazoline)s and the polymethylmethacrylate standards from the SEC analysis. This scaling procedure is strictly only valid in the low to intermediate monomer conversion range at which the presence of linear chains can be expected as good as exclusive[50]. In the present work, the factor is also suited for the higher monomer conversions, as branching is very limited (Supplementary Figs. 7-10; values below 3 mol %).

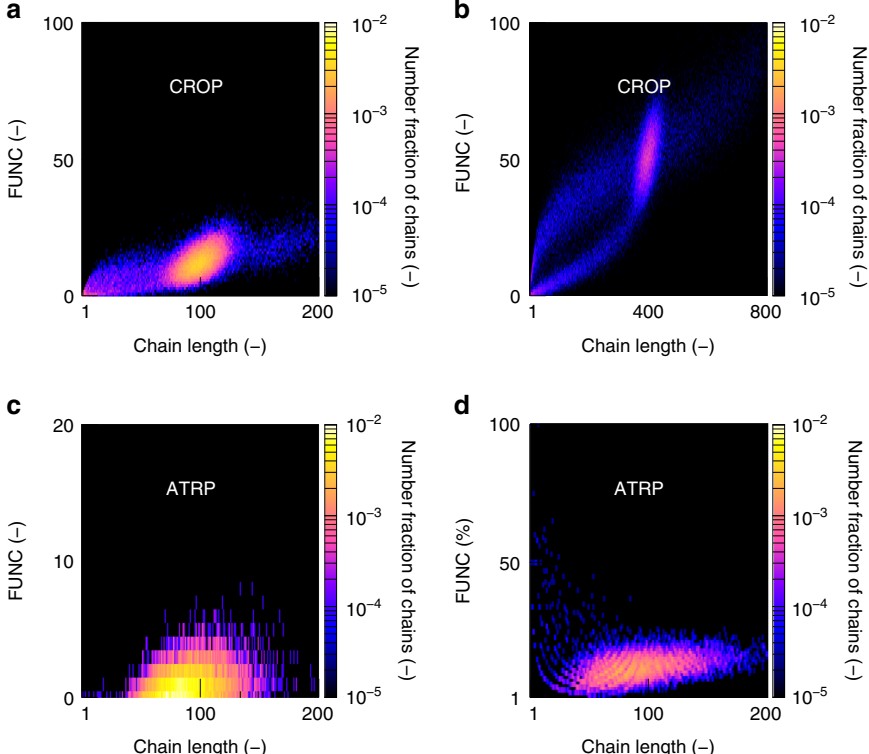

**Fig. 10** Generic nature of concept of functionality-chain length distributioons. **a**, **b** case of synthesis of functional copoly(2-oxazoline) at 100% monomer conversion for target DP of **a** 100 and **b** 400 (optimized conditions from Fig. 9). **c**, **d** case of atom transfer radical polymerization (ATRP) of **c** styrene and N-propyl maleimide; and **d** ethylhexyl acrylate and glycidyl methacrylate (monomer conversion of 90%); conditions for (**c**): target DP of 100; [styrene]$_0$: [R$_0$X]$_0$:[Activator]$_0$: 100:1:1; 110 °C; addition of 1 eq. of N-propyl maleimide at 35% styrene conversion with respect to ATRP initiator R$_0$X; conditions for (**d**): target DP of 100; [M]$_0$:[R$_0$X]$_0$: [Activator]$_0$: 100:1:1; 10 mol % of glycidyl methacrylate at start; solvent toluene; 70 °C[31,32]

For the experimental determination of the $f_{nonfunctionalized}$, PMeOx-PC2MestOx (DP = 25, $f_{C2MestOx,0}$ = 5 mol %, 200 mg) and LiOH (10 equivalent; C2MestOx groups) were dissolved in distilled water and stirred overnight, followed by purification by PD-10 column in water and freeze-drying. The obtained carboxylic acid-containing copolymer PMeOx-PC2COOHOx (100 mg) was then dissolved in degassed N,N-dimethylacetamide (5 mL). After addition of N,N′-diisopropylcarbodiimide (3 equivalent; COOH groups), triethylamine (5 equivalents; COOH groups) and a catalytic amount of 4-dimethylaminopyridine, the mixture was added into a flask containing Wang resin (2 g, 200–400 mesh, hydroxyl group content 1.7 mmol/g) pre-swollen in dichloromethane (10 mL). The reaction mixture was purged with argon and vortexed at room temperature for 4 days. Afterwards, the resin was washed repeatedly with N,N-dimethylacetamide to separate the non-functionalized (PMeOx) homopolymer, which was further purified by PD-10 column in water. The functionalized polymer was cleaved from the resin using a mixture of trifluoroacetic acid and dichloromethane (1:1, 20 mL, 5) and purified by PD-10 column in water. $f_{nonfunctionalized}$ was determined by $^1$H NMR spectroscopy according to $f_{nonfunctionalized} = 1 - (I_1/I_2)$, with $I_1$ and $I_2$ the relative integrals of the side-chain group peaks ($\delta$ = 2.7 ppm) of PMeOx-PC2COOHOx before and after the removal of the non-functionalized polymer fraction (Supplementary Fig. 27).

For the catalytic RDRP cases, i.e., the two ATRP cases, the initiator are, respectively, 1-bromoethyl benzene and methyl 2-bromo propionate, the catalysts are, respectively, Cu(I) bromide 4,4′-dinonyl-2,2′-bipyridine and Cu(I) chloride/,2′-bipyridine, and the comonomer pairs are, respectively, styrene/propyl maleimide and 2-ethylhexyl acrylate/glycidyl methacrylate. These radical polymerizations have been experimentally studied in literature[31,32], as highlighted in the Supplementary Discussion.

**Theoretical framework**. The CROP in acetonitrile is analyzed with the kinetic Monte Carlo (kMC) modeling method as previously applied by the authors for the simulation and design of the synthesis of steep gradient copolymers on the basis of MeOx and 2-phenyl-2-oxazoline (PhOx)[50]. Post-processing of kMC simulated data allows to calculate the chain length distribution (CLD) and, therefore, the absolute size exclusion chromatogram (SEC) trace, along with average characteristics such as $x_n$ and dispersity[36]. Also compositional features can be obtained, starting from the functionality-CLD (FUNC-CLD; Fig. 2).

Next to the basic chain-growth reactions, i.e., chain initiation and propagation, chain transfer (to monomer) ($k_{trM}$) leading to the formation of macromonomer ($D_i$), and macropropagation ($k_{pm}$) are accounted for in the kMC CROP model (Fig. 1), considering a terminal model for the description of the intrinsic reactivities. Such terminal model only differentiates based on the nature of the last monomer unit, which is an acceptable assumption for copolymerization processes such as CROP that have been less studied and for which no penultimate model parameters are currently available. Diffusional limitations on the reaction rates can be ignored, due to the diluted conditions and the relatively low intrinsic reactivities as displayed in Table 1 and Supplementary Table 1-4[45,47,53].

For the four comonomer pairs, the Arrhenius parameters for the CROP reactions in Fig. 1 (top) are provided in Supplementary Table 1-4. In these tables, also the rate coefficients at 140 °C are given. For chain initiation with MeOx, i.e., nucleophilic addition of MeOx to $I$, the intrinsic reactivity is taken from the recent study of Van Steenberge et al[48]. For the three other comonomers (EtOx, C2MestOx, and C3-MestOx) the chain initiation rate coefficient ($k_i$) has been tuned based on homopolymerization dispersity data at low monomer conversions (Supplementary Fig. 1-6). For the homo-propagation rate coefficients ($k_{p,ii}$) literature parameters are used[43,46,54]. In agreement with previous work[50] the homo-chain transfer coefficients ($k_{trM,ii}$) are tuned based on dispersity data at the higher monomer conversions (Supplementary Discussion).

Hence, for the CROP case, the largest number of kinetic parameters are determined based on straightforward homopolymerization data. The remaining limited number of cross-propagation/chain transfer rate coefficients ($k_{p/trM,ij}$; i ≠ j) have been optimized by a comparison with experimental copolymerization data, in particular the measured conversion profiles of the individual comonomers. To enhance the sensitivity toward parameter tuning, copolymerizations under equimolar conditions, i.e., with the same initial comonomer concentrations, are deliberately used (Supplementary Discussion).

For the ATRP cases, which are included to further illustrate the general potential of the developed research strategy, literature based parameters are used, as explained in the Supplementary Discussion.

## Data availability
The authors declare that all data supporting the findings of this study are available within the open literature, that paper and its supplementary information files.

## Code availability

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

## Acknowledgements

P.H.M.V.S. acknowledges the research foundation—Flanders (FWO) through a post-doctoral fellowship. B.V. acknowledges support from the Agency for Innovation by Science and Technology (IWT). R.H. is grateful to the Special Research Fund of Ghent University (BOF-UGent) and the FWO for funding. The authors also thank the Inter-university Attraction Poles Program–Belgian State–Belgian Science Policy for financial support. D.R.D. acknowledges the FWO.

## Author contributions

P.H.M.V.S., J.C.H.O., M.F.R. and D.R.D. contributed to the kinetic modeling part. B.V., O.S. and R.H. contributed to the experimental part. All authors have approved the manuscript and made significant revisions along its preparation.

## Additional information

**Competing interests:** R.H. is one of the founders of Avroxa BVBA that commercializes poly(2-oxazoline)s as Ultroxa®. The remaining authors declare no competing interests.

