## [Peer Review File · Nature Communications]

Reviewers' comments:

Reviewer #1 (Remarks to the Author):

This manuscript presents a model constructed to describe a large set of previously published data for the copolymerization of 2-oxazoline via catalytic ring-opening polymerization (CROP). The model is validated against experimentally-determined monomer and comonomer conversions and polymer average chain-length and dispersity determined over a range of initial conditions for batch operation. The model is then used to illustrate the extent of composition drift that occurs with conversion. The kinetic Monte Carlo (KMC) technique utilized also provides predictions of how the functional monomers (present at less than 20%) are distributed among the polymer chains produced. It is shown that this latter distribution is significantly influenced by re-incorporation/reactivation of copolymer chains formed earlier in the reaction via a chain transfer mechanism.

Visualization of the composition drift is provided via a novel "fingerprint" representation (Figure 5). While the predicted bimodal and trimodal fingerprints are not verified experimentally, they are a logical result from the carefully-developed model. The manuscript then defines an overall measure of the "functionalization quality", as well as reporting the amount of non-functionalized material. This measure may be useful if it can be correlated to an experimental determination of copolymer quality; however, this is not attempted in the current study.

A series of simulations are then conducted to illustrate how the distribution of functional groups is influenced by initial fraction of functional monomer in the recipe and target chain length (Figure 8). These results are not surprising to those familiar with copolymerization, except for the effect of secondary reactions (already demonstrated by Figure 5). Temperature has little effect (supplemental information), while the relative reactivity of the comonomer is important (certainly not a surprise). This leads to Table 1, which summarizes the findings from the complete set of simulations.

The manuscript is well written, and provides a complete story of model development and verification, and how the model might be utilized to produce copolymer with a more uniform distribution. Minor questions are:

1. According to the experimental data in Figure 3c, there is no significant variation in dispersity with reaction temperature. The accompanying text, based upon the simulation results shown, argue otherwise; i.e., that dispersity increases with increasing temperature. This discussion should be modified, as it is not correct to make this assertion without experimental evidence.
2. The discussion on page 8 should refer to Figure 4, not Figure 3.
3. It would be helpful to include the color coding of Figures S27 in the main text.

Significance of work: The issue of comonomer composition drift in a batch reaction is well-known, as is the importance of the distribution of functional comonomer units. The KMC technique has previously been applied to model this distribution in radical copolymerization systems, and to demonstrate the difficulty of maintaining uniform distribution among all chains as the target chain length in the system is decreased. The specific results from the series of simulations summarized in Table 1 are perhaps useful for research groups working specifically with poly(2-oxazoline) systems, but may not be of interest to a larger audience. Not mentioned at all, however, is the obvious answer of how the uniform distribution of functional groups can be improved, which is through semibatch operation (as used industrially).

Aspects that are new to this work are (a) a full and verified model of the CROP chemistry; (b) application of KMC techniques to examine copolymer composition distribution for functionalized 2-oxazoline copolymers; (b) use of the model to demonstrate that functional group distribution can be strongly affected by side reactions; (c) development of tools to visualize the copolymer composition distribution and provide an overall "functionalization quality index" (although not compared to experiment).

I consider this a very nice combination of model development and advanced simulation techniques applied to a particular copolymerization chemistry that requires only minor revision. I would recommend it for publication in a top polymer chemistry (such as Polym. Chem. or

Macromolecules) or engineering journal without any reservations. However, I am not certain whether the work is of high enough significance to appear in Nature Commun, a decision I leave to the editor.

Reviewer #2 (Remarks to the Author):

The authors present a detailed study of the kinetics of several oxazoline copolymerizations, taking into account the major side reaction of chain transfer to monomer. They have constructed a kinetic Monte Carlo model for the polymerization, which they use to determine rate constants for many of the elementary reactions by fitting the model output to experimental data (conversion, number average molecular weight, dispersity). The refined model is then used to generate a bivariate distribution of degree of polymerization and composition (or functionalization). This approach reveals that at high conversion, the DP/functionalization distribution is trimodal, containing fractions of low MW, low functionality and low MW, high functionality polymer in addition to the main peak. Finally, the authors demonstrate that the experimental conditions can be adjusted to reduce the breadth of the functionality distribution and the amount of low functionality polymer in a polymerization containing 13% functional oxazoline.

In my opinion, this is a solid piece of work from a group that has an exceptional track record in this area. The main strength of the article is that it draws attention to the distribution of composition that may be encountered in a copolymerization, such that a significant fraction of chains may deviate from the average composition. This is poorly appreciated in the polymerization community, and the authors are among the few researchers calling attention to this point.

In its current form, however, I think the article is too specialized to be suitable for publication in Nature Communications. This is because of the following points:

- * it is very specific to oxazoline polymerization. Although the authors state that their technique is generic and extendable to a broad range of copolymerization technologies, only oxazoline-based examples are given. The paper would have much greater impact if an alternate form of polymerization were also modelled using the technique (e.g. an RDRP or ROMP).
- * There is no experimental validation of the distribution of functionalization. Admittedly, this would be difficult to do, but how can the reader be confident that the results of the model are correct? Is it possible to measure at least the fraction of non-functionalized polymer? My concern is that the functionalization distribution is determined using a model that contains a very large number of parameters, none of which are precisely known. How does the uncertainty in the parameters affect the precision of the output?
- * Following up on this point, the agreement between the experimental data and the model is sometimes poor - particularly in the case of dispersity data. See for example Figure 3 (blue triangles), Figure S3 (yellow triangles), Figure S6 (yellow circles). In other cases, the model is consistent with the data, but the data are quite imprecise. If I have understood the method correctly, the chain transfer constants were obtained largely by fitting the dispersity data at high conversions. Given the uncertainty in this data, what uncertainty is associated with the chain transfer constants and how does this affect the functionality distribution?
- * It is not very clear how the chain transfer constants were determined. I looked also at Macromolecules 48, 7765-7773 which was the source for one of the chain transfer constants, and it was not very clear in that reference either. My impression is that the chain transfer constants were adjusted until the output of the model matched that of the experimental data. These parameters seem quite crucial to the final functionality distribution, as chain transfer is the main side reaction in the polymerization. Determining 4 separate chain transfer constants from a single set of experimental data does not seem at all trivial, and I think it would be worth going into significantly more detail about how this was done.

Some minor points:

*The classification of functionalization quality seems a bit arbitrary - how were the cutoff levels for excellent, good, moderate and bad determined?

* p8 l170, text refers to Figure 3, but should be fig 4

* FUNC describes the number of functional groups per chain - this is highly correlated with chain length as longer chains tend to have a greater number of functional groups. It could be interesting to look at the proportion of functional groups relative to the total number of monomers rather than the number of functional groups.

In summary, I think this is a very strong modelling paper that would be very suitable for a more specialized journal, but in its current form it is too focused on a single polymerization technique for publication in Nature Communications. In addition, I have some concerns regarding the accuracy of the parameters used in the model and the effect that this might have on the conclusions which should be addressed.

Answers to comments of reviewer #1

General analysis

This manuscript presents a model constructed to describe a large set of previously published data for the copolymerization of 2-oxazoline via catalytic ring-opening polymerization (CROP). The model is validated against experimentally-determined monomer and comonomer conversions and polymer average chain-length and dispersity determined over a range of initial conditions for batch operation. The model is then used to illustrate the extent of composition drift that occurs with conversion. The kinetic Monte Carlo (KMC) technique utilized also provides predictions of how the functional monomers (present at less than 20%) are distributed among the polymer chains produced. It is shown that this latter distribution is significantly influenced by re-incorporation/reactivation of copolymer chains formed earlier in the reaction via a chain transfer mechanism.

Visualization of the composition drift is provided via a novel “fingerprint” representation (Figure 5). While the predicted bimodal and trimodal fingerprints are not verified experimentally, they are a logical result from the carefully-developed model. The manuscript then defines an overall measure of the “functionalization quality”, as well as reporting the amount of non-functionalized material. This measure may be useful if it can be correlated to an experimental determination of copolymer quality; however, this is not attempted in the current study.

A series of simulations are then conducted to illustrate how the distribution of functional groups is influenced by initial fraction of functional monomer in the recipe and target chain length (Figure 8). These results are not surprising to those familiar with copolymerization, except for the effect of secondary reactions (already demonstrated by Figure 5). Temperature has little effect (supplemental information), while the relative reactivity of the comonomer is important (certainly not a surprise). This leads to Table 1, which summarizes the findings from the complete set of simulations. The manuscript is well written, and provides a complete story of model development and verification, and how the model might be utilized to produce copolymer with a more uniform distribution.

Answer to the general analysis

We thank the reviewer for his or her general appreciation of the work. Indeed the current work puts forward the use of novel fingerprints, i.e. functionality – chain length distributions (FUNC-CLDs), to enable an unbiased assessment of the functionalization quality, by focusing both on the fraction of non-functional chains and the diversity in the functionalization degree per chain length. The consideration of side reactions is essential, enabling the use of the modeling tool to also identify optimal synthesis conditions in view of excellent functionalization behavior.

It should be stressed that the current work illustrates how modeling and experimental analysis can be synergistically combined to obtain reliable FUNC-CLDs. Three steps are performed. A first step is the identification of a detailed reaction scheme which is not restricted to main reactions but also accounts for side reactions. For the CROP case, as recognized by the reviewer, this relates to chain transfer reactions. A second step is the determination of the chemical reactivities based on experimentally accessible responses such as the (co)monomer conversion profiles, the number average chain length and the dispersity. In the present work, homopolymerization data are used to determine most of the kinetic parameters. The remaining limited number of kinetic parameters follow from copolymerization data where deliberately equimolar conditions are considered to allow

for sufficient parameter sensitivity. As illustrated in the Supporting Information the agreement between model and experimental data is acceptable, taking into account unavoidable experimental error.

The third step is the full utilization of a powerful modeling tool in which chain-to-chain deviations can be depicted by employing efficient storage and search algorithms. In the present work, an advanced kinetic Monte Carlo algorithm is considered to be successful in this respect. Hence, provided that reliable kinetic parameters have been determined in the previous step, detailed information on the molecular level that is almost inaccessible with experimental tools becomes available, which is highly relevant for the research field. Post-processing allows to obtain all desired characteristics such as the number of chains that are not functionalized. We thus put forward that the correct execution of step 1 to 3 automatically results in the correct mapping of the functionalization pattern over the chain lengths. This is consistent with the interpretation of the reviewer that the obtained results with this last step are logical.

The reviewer is correct that further experimental validation – so related to the third step – can be useful to further illustrate the correctness of the last step. In the original work this was not attempted, as highlighted by the reviewer. In the revised manuscript, we have included new data that demonstrate that the experimentally determined fraction of chains that are not functionalized is in agreement with the simulated data. Focus is here on conditions at which this fraction is very high to enable experimental detection.

Minor comment 1

According to the experimental data in Figure 3c, there is no significant variation in dispersity with reaction temperature. The accompanying text, based upon the simulation results shown, argue otherwise; i.e., that dispersity increases with increasing temperature. This discussion should be modified, as it is not correct to make this assertion without experimental evidence.

Answer to minor comment 1

The reviewer is correct that the comparison between experimental and modeled data in Figure 3c is not perfect. In particular, the data at 100°C (blue points) are experimentally higher. Note that the experimental trend for the other three temperatures (80, 120 and 140 °C) is grasped by the kinetic model and the overall description of the experimental data in Figure 3c is still acceptable.

More importantly is the evaluation of the complete data set beyond Figure 3c, implying the overall analysis of Figure S1-S6 in the Supporting Information. It follows that the data are described in an acceptable manner.

It should be further stressed that a kinetic model – once based on a sufficient broad set of experimental data – can filter out unrealistic parameter combinations and identify less reliable experimental data. Arrhenius behavior can be expected for individual reaction steps as correctly implemented in the kinetic model. As chain transfer is the contributor to the dispersity increase and the relevant importance of this reaction increases at higher temperature, in agreement with literature data, it is thus understandable why the optimized model predicts lower dispersities for the 100°C data in Figure 3d. This again illustrated the strength of a combined experimental and modeling approach.

This aspect has been made clear in the revised manuscript.

Minor comment 2

The discussion on page 8 should refer to Figure 4, not Figure 3.

Answer to minor comment 2

The reviewer is correct and the update has been made.

Minor comment 3

It would be helpful to include the color coding of Figures S27 in the main text.

Answer to minor comment 3

This indeed increases the readability as this color coding is relevant to grasp the differences in functionality quality, with a gradient from bad (red) to excellent (dark green). The color coding has therefore be incorporated as a subfigure in Figure 7 (Figure 7(d)).

General comment 1

The issue of comonomer composition drift in a batch reaction is well-known, as is the importance of the distribution of functional comonomer units. The KMC technique has previously been applied to model this distribution in radical copolymerization systems, and to demonstrate the difficulty of maintaining uniform distribution among all chains as the target chain length in the system is decreased. The specific results from the series of simulations summarized in Table 1 are perhaps useful for research groups working specifically with poly(2-oxazoline) systems, but may not be of interest to a larger audience. Not mentioned at all, however, is the obvious answer of how the uniform distribution of functional groups can be improved, which is through semibatch operation (as used industrially).

Answer to general comment 1

The reviewer is correct that semibatch operation, which typically involves the addition of one of the comonomers with increasing time as opposed to the batch case with all comonomer present at the start, has been conducted in an industrial context. Essential is here the so-called Mayo-Lewis plot as also included in the present work (Figure 4).

It should however be stressed that this relates to conventional chain growth polymerization (e.g. free radical polymerization) which does not allow the synthesis of the advanced well-defined polymer products as targeted in the present work. Here focus is on low dispersity well-defined products for advanced applications in the field of polymer therapeutics and hydrogels aiming at full monomer conversion, hence, batch conditions. Moreover, as illustrated in Figure 4, the Mayo-Lewis shape is rather unsuited for semibatch procedures and the key disturber for chain growth control is chain transfer. This aspect has been made clear in the revised manuscript.

The relevance of the present work has also been further illustrated for reversible deactivation radical polymerization (RDRP), with the main results included in an updated Figure 9.

General comment 2

Aspects that are new to this work are (a) a full and verified model of the CROP chemistry; (b) application of KMC techniques to examine copolymer composition distribution for functionalized 2-oxazoline copolymers; (b) use of the model to demonstrate that functional group distribution can be strongly affected by side reactions; (c) development of tools to visualize the copolymer composition distribution and provide an overall “functionalization quality index” (although not compared to experiment).

Answer to general comment 2

We thank the reviewer for highlighting the novel aspects of the work. In the revised manuscript the application has been also extended from CROP to RDRP. As explained above a correct execution of the strength of model and experimental analysis allows to obtain reliable FUNC-CDLs.

Conclusion

I consider this a very nice combination of model development and advanced simulation techniques applied to a particular copolymerization chemistry that requires only minor revision. I would recommend it for publication in a top polymer chemistry (such as Polym. Chem. or Macromolecules) or engineering journal without any reservations. However, I am not certain whether the work is of high enough significance to appear in Nature Commun, a decision I leave to the editor.

Answer to conclusion

We again greatly thank the reviewer for his or her recognition of the quality of the work as such. Indeed, the original manuscript focused only on one copolymerization chemistry. In the revised manuscript, as also suggested by the editor, we have also included the application to RDRP. Furthermore, the developed model is validated by experimental validation of the fraction of chains that is non-functional.

Answers to comments of reviewer #2

General analysis

The authors present a detailed study of the kinetics of several oxazoline copolymerizations, taking into account the major side reaction of chain transfer to monomer. They have constructed a kinetic Monte Carlo model for the polymerization, which they use to determine rate constants for many of the elementary reactions by fitting the model output to experimental data (conversion, number average molecular weight, dispersity). The refined model is then used to generate a bivariate distribution of degree of polymerization and composition (or functionalization). This approach reveals that at high conversion, the DP/functionalization distribution is trimodal, containing fractions of low MW, low functionality and low MW, high functionality polymer in addition to the main peak. Finally, the authors demonstrate that the experimental conditions can be adjusted to reduce the breadth of the functionality distribution and the amount of low functionality polymer in a polymerization containing 13% functional oxazoline.

In my opinion, this is a solid piece of work from a group that has an exceptional track record in this area. The main strength of the article is that it draws attention to the distribution of composition that may be encountered in a copolymerization, such that a significant fraction of chains may deviate from the average composition. This is poorly appreciated in the polymerization community, and the authors are among the few researchers calling attention to this point. In its current form, however, I think the article is too specialized to be suitable for publication in Nature Communications. This is because of the following points:

Answer to general analysis

We thank the reviewer for his or her general appreciation and also the nice words on the track record. Indeed the aspect of chain-to-chain deviations is not truly recognized in the research field. It is correct that the original manuscript focused on one experimental technique. This has been resolved in the revised manuscript by extending the application to also reversible deactivation radical polymerization (RDRP), including two case studies with different comonomer incorporation reactivities. We are convinced that by doing so the manuscript is now accessible to a broader audience as it now explicitly highlights its generic character. Also we have covered the additional points of the reviewer, as explained below.

General comment 1

it is very specific to oxazoline polymerization. Although the authors state that their technique is generic and extendable to a broad range of copolymerization technologies, only oxazoline-based examples are given. The paper would have much greater impact if an alternate form of polymerization were also modelled using the technique (e.g. an RDRP or ROMP).

Answer to general comment 1

We thank the reviewer for this suggestion. In the revised manuscript, we have included results on RDRP, selecting atom transfer radical polymerization (ATRP) as reference technique. Focus is on ATRP of styrene with small amounts of N-propyl maleimide and ATRP of 2-ethylhexyl acrylate and glycidyl

methacrylate. The main new figure is Figure 9, with the details on the ATRP model parameters and validation provided in a novel Section S13 in the Supporting Information.

General comment 2

There is no experimental validation of the distribution of functionalization. Admittedly, this would be difficult to do, but how can the reader be confident that the results of the model are correct? Is it possible to measure at least the fraction of non-functionalized polymer? My concern is that the functionalization distribution is determined using a model that contains a very large number of parameters, none of which are precisely known. How does the uncertainty in the parameters affect the precision of the output?

Answer to general comment 2

It should be stressed that the current work illustrates how modeling and experimental analysis can be synergistically combined to obtain reliable functionality - chain length distributions (FUNC-CLDs) that are the basis for the further quality assessment with derived characteristics such as the functionality distribution (FUNC_D), the fraction of non-functionalized chains (NONFUNC value) and the variation in the functionalization degree ($C_{V,FUNC}$ value).

The methodology is constructed around three steps. A first step is the identification of a detailed reaction scheme which is not restricted to main reactions but also accounts for side reactions. A second step is the determination of the chemical reactivities based on experimentally accessible responses such as the (co)monomer conversion profiles, the number average chain length and the dispersity. Indeed a large number of parameters need to be determined but this can be fortunately done in a stepwise manner, as better explained in the revised manuscript. Homopolymerization data are used to determine most of the kinetic parameters. The remaining limited number of kinetic parameters follow from copolymerization data where deliberately equimolar conditions are considered to allow for sufficient parameter sensitivity. As illustrated in the Supporting Information the agreement between model and experimental data is acceptable, taking into account unavoidable experimental error. The third step is the full utilization of a powerful modeling tool in which chain-to-chain deviations can be automatically depicted by employing efficient storage and search algorithms. Provided that dedicated parameter tuning is performed in step a reliable functionalization distribution thus results.

Additional validation is always useful but far from trivial, again highlighting the relevance of the present work. To further increase the quality of the manuscript several experimental analysis techniques have been considered to obtain fraction of chains that are not functionalized (NONFUNC value). In the revised manuscript, it has been demonstrated that that the experimental NONFUNC is in agreement with the simulated data. Focus is here on conditions at which this fraction is very high to enable experimental detection.

General comment 3

Following up on this point, the agreement between the experimental data and the model is sometimes poor - particularly in the case of dispersity data. See for example Figure 3 (blue triangles), Figure S3 (yellow triangles), Figure S6 (yellow circles). In other cases, the model is consistent with the data, but the data are quite imprecise. If I have understood the method correctly, the chain transfer constants were obtained largely by fitting the dispersity data at high conversions. Given the uncertainty in this data, what uncertainty is associated with the chain transfer constants and how does this affect the functionality distribution?

It is not very clear how the chain transfer constants were determined. I looked also at Macromolecules 48, 7765-7773 which was the source for one of the chain transfer constants, and it was not very clear in that reference either. My impression is that the chain transfer constants were adjusted until the output of the model matched that of the experimental data. These parameters seem quite crucial to the final functionality distribution, as chain transfer is the main side reaction in the polymerization. Determining 4 separate chain transfer constants from a single set of experimental data does not seem at all trivial, and I think it would be worth going into significantly more detail about how this was done.

Answer to general comment 3

The reviewer is correct that the not all experimental data are described to the same extend by the model. It should however be stressed that a comparison of experimental and modeled data needs to be done globally. For the CROP case study, this implies an inspection of Figure S1-6 in the Supporting Information. It follows – in agreement with the comment of the reviewer – that on an overall basis the agreement is acceptable.

Moreover, the model can be used to filter out experimental error and to identify less correct experimental data. Based on previous work it is clear that dispersity increases are associated with chain transfer. As Arrhenius behavior is expected for the chain transfer reaction it is to be expected to low temperature data are less affected by this reaction. Globally this is indeed the case but in some experiments, likely due to too high experimental error, this is not that case. This leads at first sight to a biased interpretation but due to the combination of experimental and modeling research one is capable to correct for such phenomena during the parameter optimization. The theoretical screening thus strongly helps toward the reliable parameter determination.

The latter is specifically true for the dispersity data. Again due to the differentiation between homo and copolymerization data it is possible to maximize the reliability toward parameter determination and in particular chain transfer.

Specifically the modeling methodology has been improved to make the reader aware of this parameter tuning strategy, as suggested by the reviewer.

Minor comment 1

The classification of functionalization quality seems a bit arbitrary - how were the cutoff levels for excellent, good, moderate and bad determined?

Answer to minor comment 1

The reviewer is correct that the classification can be seen as somewhat arbitrary. The major goal is to have a transition from an excellent to a very bad functionalization quality. The boundary NONFUNC values are determined based on expected requests from the application side. The critical $C_{V, FUNCD}$ value of 0.5 is selected based on literature data involving distributed properties. Note that the $C_{V, FUNCD}$ and NONFUNC boundaries at least allow a ranking of functional copolymers impossible based on experimental research only. Moreover, the color trends are relevant for application as purification by preparative size exclusion chromatography is impossible and preparative liquid chromatography for polymers is cumbersome. These aspects have been also made clear in the revised manuscript.

Minor comment 2

p8 l170, text refers to Figure 3, but should be fig 4

Answer to minor comment 2

The reviewer is correct and the update has been made.

Minor comment 3

FUNC describes the number of functional groups per chain - this is highly correlated with chain length as longer chains tend to have a greater number of functional groups. It could be interesting to look at the proportion of functional groups relative to the total number of monomers rather than the number of functional groups.

Answer to minor comment 3

We are somewhat puzzled by this comment as the FUNC-CLD, which is the key property in our work, explicitly covers this correlation as mentioned by the reviewer. The FUNC-CLD highlight the fraction of chains with a given chain length and a given functional comonomer amount. Hence, inherently for both low and high chain length a differentiation is made. A key figure in the manuscript is Figure 5 depicting the monomer conversion dependency of FUNC-CLD and the relevance of the initial functional comonomer amount. Scatter in both the x and y direction are observed supporting the interpretation of the reviewer.

It should be stressed that based on these bivariate FUNC-CLDs derived properties such as the functionality distribution are calculated to allow for an unbiased qualification. At any point the original FUNC-CLDs are thus available.

Conclusions

In summary, I think this is a very strong modelling paper that would be very suitable for a more specialized journal, but in its current form it is too focused on a single polymerization technique for publication in Nature Communications. In addition, I have some concerns regarding the accuracy of the parameters used in the model and the effect that this might have on the conclusions which should be addressed.

Answer to conclusions

We highly appreciate that the reviewer acknowledges the strength of the work. It is correct that in the original manuscript the focus was on one copolymerization technique. In the revised manuscript, as explained above we have highlighted that the combined experimental and modeling methodology can also be applied for e.g. reversible deactivation radical polymerization, as also suggested by the editor. Moreover, in the revised manuscript, we have also better explained that the kinetic parameters determination as originally conducted is sufficiently reliable in view of the application of the model for design of functional copolymers.

Reviewers' comments:

Reviewer #1 (Remarks to the Author):

The experimental determination of non-functional chains to validate model predictions shown in Figure 8b, and the ATRP examples presented as Figures 9c-d are valuable additions to an excellent manuscript. (See previous review.) I recommend acceptance after attending to the following list of minor revisions.

Line 135 Sentence fragment: "The chains are although functionalized (FUNC > 0)." should be removed.

Line 224-226: Shouldn't this discussion refer to Figure 4c (not 3c)?

Figure 5 caption: why is information regarding Figure S15 provided here? This should only be discussed in the main body of the text.

Line 332: Should be "It is thus again shown that..." rather than "It is this again showed that..."

Line 450: Better to say "and low initial loadings of functional monomer"

Line 460 states a calculated non-functional fraction of 32%, while 31% is marked on Figure 8b.

Line 509-514: This paragraph needs to be rewritten to improve clarity: the current phrasing is awkward, making the central premise of the discussion difficult to understand.

Figure 9: It would be useful to label the c and d plots with "ATRP" (similar to "CROP" shown for a and b). In addition, I suggest that the text in parentheses – "(dark green color; Figure S27 and Figure 7(d))" – be removed from the caption: I find it confusing, as there is no green included in the scale for Figure 9. I understand why the authors added it, but find it unnecessary.

Line 570: I find the statement "addition of 1 equivalent with respect to the initial styrene amount" confusing. Does this mean 50 mol%? Please clarify.

Line 581: Better to say "chain-to-chain deviations are still identified that can be attributed" (rather than "can be still...")

Reviewer #2 (Remarks to the Author):

The authors have made quite substantial revisions to their manuscript, especially by including some results on ATRP, thus demonstrating that their model can be generalized to other polymerization techniques. I think this is a good quality article on an important topic, and while I still have a number of mostly technical comments, I think that with suitable revision this manuscript could be accepted for publication in Nature Communications.

As a general point of style, the manuscript is quite difficult to read, in part because the reader is frequently directed to the Supporting Information. Where possible, this should be avoided. Additionally, there is a proliferation of similarly named functions and distributions: FUNC-CLD, FUNC, FUNCD, <FUNCinst>, <FUNC>, NONFUNC... it is hard to keep them all straight, especially when their definitions change (FUNC is the number of functionalities per chain, except when it is equal to 1-NONFUNC in Figure S16).

As <FUNCinst> is equivalent to $F_{(A,inst)}$, why not use $F_{A,inst}$ throughout? Likewise, <FUNC> could be redesignated $F_{(A,cum)}$, following standard polymer notation.

X_m is used throughout the text as an abbreviation for (total) monomer conversion - it would be easier to read if this were spelled out.

p2. "the synthesis of functional polymers is commonly performed..." this makes it sound as CROP and RDRP are the main techniques for functional polymer synthesis, it would be better to rephrase so that it is clear that these are two possibilities out of many.

p6. I don't think x_n has been defined yet. Beware of possible confusion with X_m.

Figure 2 caption: 'most chains have a length of 100 and FUNC of 50' it would be more correct to say the most probable chain length and FUNC is 100 and 50 (i.e. this is the mode of the distribution).

In Figure 3 and in Figures S1-S15, how was the size of the error bars determined? Do they represent the standard error, or a confidence interval? This should be specified.

The discrepancy between experimental and theoretical dispersities is real, and requires some comment - while the observed dispersities are broadly consistent with the predictions, the effect of temperature on dispersity that is predicted by the model is simply not observed. Perhaps this is due to the difficulty of obtaining accurate values, or perhaps the model is wrong - in any case, the discrepancy must be noted and some attempt should be made to explain it.

I don't think Figures S13 and S14 are necessary. The additional information provided by the figures is offset by the inconvenience of having to search for them in the Supporting Information. That the copolymer composition is different to the comonomer feed is well-understood, and I don't really see how it shows that the polymerization should not be stopped at lower conversion - frequently in this situation one runs the polymerization at a monomer feed that gives the desired copolymer composition, then stops the reaction before significant composition drift takes place (e.g. around 75% in this case). Also Fig S13 shows <FUNC> nearly always greater than f_(C2MestOx), while Fig 4c shows <FUNCinst> always < f_(C2MestOx) - how is this possible? Regarding Fig S14, it could simply be stated that an identical composition profile is obtained in the absence of chain transfer - this is not particularly surprising.

On p12, the discussion of the origins of the trimodal fingerprint could refer to Figure 5 (e.g. the lower arm is due to chain transfer, the top arm to chain initiation after chain transfer). The reader could then be referred to Figure S15 (e.g. "this is illustrated in Figure S15, in which different side reactions are successively added to the polymerizations"). The idea is to avoid having to continually refer to the supporting information in order to understand the article.

p14: "It should be stressed..." I don't think there's any need for this sentence, which appears meant for the reviewers/editor more than for the reader.

Figure 6(c) looks wrong to me: the black distribution in Figure 6a roughly corresponds to a Poisson with mean 0.7, while the black Poisson distribution in 6c has a mean of roughly 1.5. But the means of the two distributions are stated to be the same. Maybe the authors could check this.

As Figure S27 duplicates Fig 7d, Fig S27 should be deleted, and all references in the text should be to Fig 7d. I still find the descriptions of functionalization quality entirely arbitrary and the characterization from "bad" to "excellent" seems unjustified. What is special about < 2% non-functional chains and C_v < 0.4? At the least, these criteria should be explained in the text (e.g. the text accompanying Fig S27 should be moved to the main article).

In the subsequent discussion, distributions are repeatedly described by their color or whether they are good or bad, and these arbitrary distinctions are thus invested with spurious meaning. Why is the difference between entries 9 and 10 of Table 9 significant, but the difference between entries

10 and 6 is not? I feel like these arbitrary categories make it more difficult to understand the results, not less.

On p18, the authors have inserted "the latter" - it's not clear if this refers to the too high Cv or the 'excellent' NON-FUNC.

On p19, it would be better to say "they are less likely to undergo chain transfer" instead of 'they are more likely to not undergo chain transfer'. The authors then state that the calculation of an average becomes less-defined for lower target DPs. The calculation of the average is still well-defined, perhaps they mean the concept of the average becomes less meaningful?

On p21, it is great that the authors have been able to test their conclusions experimentally. I would note that if the number of functionalities per chain were Poisson distributed, one would expect $e^{-1.25} = 28.7\%$ of chains to be non-functionalized in this case - an agreement that is even better (more excellent?) than that of the model. On this point, I think it would be better to say that the experiment returned a result that was within 4% of the prediction, rather than using loaded terms like 'excellent'.

I have attached a graph that I suggest the authors use in place of Table 9, showing graphically the variation of Cv and NONFUNC as DP, f, monomer and T are varied. A graphical presentation makes the trends quite obvious, and erases the arbitrary excellent/very good/good/bad distinctions. Whatever the authors decide to do, I feel that the data should be discussed in terms of trends and effects of changing parameters, not in terms of whether or not they conform to an arbitrary goal.

The statement on semi-batch polymerizations that has been added to p23 needs some context in order to be understood by a typical reader - why would semi-batch procedures be of interest?

In conclusion, I think this article could be accepted for publication, but still needs some fairly substantial revision, particularly concerning the discussion of functionalization quality. Removing the discussion of 'excellence' and replacing it with a simple explanation of what factors affect Cv and NONFUNC would make the article both shorter and better. In my opinion the value of this article lies less in presenting a detailed explanation of how to make a particular functional polymer to an arbitrary quality standard and more in its presentation of the FUNC-CLD distribution and how different parameters affect Cv and NONFUNC.

Simon Harrison

Answers to comments of reviewer 1

General comment 1

The experimental determination of non-functional chains to validate model predictions shown in Figure 8b, and the ATRP examples presented as Figures 9c-d are valuable additions to an excellent manuscript. (See previous review.) I recommend acceptance after attending to the following list of minor revisions.

Answer to general comment 1

We thank the reviewer for appreciating the additional effort regarding the experimental validation and the consideration of ATRP examples. We also further thanks this reviewer for his or her effort in the review process.

Specific comment 1

Line 135 Sentence fragment: “The chains are although functionalized (FUNC > 0).” should be removed.

Answer to specific comment 1

As suggested by the reviewer, this fragment has been removed.

Specific comment 2

Line 224-226: Shouldn't this discussion refer to Figure 4c (not 3c)?

Answer to specific comment 2

Indeed the reviewer is correct. This has been corrected.

Specific comment 3

Figure 5 caption: why is information regarding Figure S15 provided here? This should only be discussed in the main body of the text.

Answer to specific comment 3

We have updated the caption as suggested by the reviewer. The discussion is thus limited to the main body of the text.

Specific comment 4

Line 332: Should be “It is thus again shown that...” rather than “It is this again showed that...”

Answer to specific comment 4

We have made this change in the revised manuscript.

Specific comment 5

Line 450: Better to say “and low initial loadings of functional monomer”

Answer to specific comment 5

Indeed, this has been changed in the revised manuscript.

Specific comment 6

Line 460 states a calculated non-functional fraction of 32%, while 31% is marked on Figure 8b.

Answer to specific comment 6

We thank the reviewer for noting this. Indeed, it is 31%. This typographical error has been resolved.

Specific comment 7

Line 509-514: This paragraph needs to be rewritten to improve clarity: the current phrasing is awkward, making the central premise of the discussion difficult to understand.

Answer to specific comment 7

We have rewritten this part to highlight more the central premise as suggested by the reviewer.

Specific comment 8

Figure 9: It would be useful to label the c and d plots with “ATRP” (similar to “CROP” shown for a and b). In addition, I suggest that the text in parentheses – “(dark green color; Figure S27 and Figure 7(d))” – be removed from the caption: I find it confusing, as there is no green included in the scale for Figure 9. I understand why the authors added it, but find it unnecessary.

Answer to specific comment 8

We have included the labeling “ATRP” in the revised manuscript, as indeed this is on the same level as “CROP”. We understand the suggestion of the reviewer regarding the caption. We have removed this part.

Specific comment 9

Line 570: I find the statement “addition of 1 equivalent with respect to the initial styrene amount” confusing. Does this mean 50 mol%? Please clarify.

Answer to specific comment 9

We thank the reviewer for noting this. This must be 1 equivalent with respect to the ATRP initiator. This aspect has been resolved in the revised manuscript.

Specific comment 10

Line 581: Better to say “chain-to-chain deviations are still identified that can be attributed” (rather than “can be still...”)

Answer to specific comment 10

We thank the reviewer for noting this. We have adapted this part.

Answers to comments of reviewer 2

General comment 1

The authors have made quite substantial revisions to their manuscript, especially by including some results on ATRP, thus demonstrating that their model can be generalized to other polymerization techniques. I think this is a good quality article on an important topic, and while I still have a number of mostly technical comments, I think that with suitable revision this manuscript could be accepted for publication in Nature Communications.

Answer to general comment 1

We thank the reviewer for his general appreciation. As explained below, we have covered the more technical comments and suggestions in the revised manuscript. We would like to also thank this reviewer for his effort in the whole review process.

General comment 2

As a general point of style, the manuscript is quite difficult to read, in part because the reader is frequently directed to the Supporting Information. Where possible, this should be avoided. Additionally, there is a proliferation of similarly named functions and distributions: FUNC-CLD, FUNC, FUNC_D, <FUNC_{inst}>, <FUNC>, NONFUNC... it is hard to keep them all straight, especially when their definitions change (FUNC is the number of functionalities per chain, except when it is equal to 1-NONFUNC in Figure S16).

Answer to general comment 2

The reviewer is correct that many links to the Supporting Information are made. We have reduced the number of times “Supporting Information” is included in the main text, when found appropriate. As such we would like to put forward that such links are unavoidable taking into the restriction regarding the total number of figures and tables in the main text.

We have also made sure that there is only one meaning for each abbreviation/symbol. This implies that the y-axis for Figure S16c is altered from “FUNC” to “f_functionalized chains” (with “_” highlighting subscript notation here). In general we have also introduced the symbol “f” to stress a fraction. Hence, NONFUNC is altered into “f_nonfunctionalized”. We thank the reviewer for noting this, as largely increases the readability.

FUNC-CLD is the core bivariate distribution based on the variates FUNC (the number of functionalities, as highlighted by the reviewer) and CL (chain length). The related distributions by integrating over respectively FUNC and CL are FUNC_D and CLD. Hence, these notations are logical but to make this more clear this definition aspect has been explicitly highlighted in the revised manuscript. Hence, we use “D” at the end to make the reader aware of the distributed nature of the variate(s) in front.

Remaining are <FUNC_{inst}> and <FUNC>. Indeed, these notations can be confusing as this is an average based on all chains either instantaneous or cumulative. We agree with the reviewer to switch this to the more conventional notation (see also the answer to specific comment 1). We now use “F” as the main symbol with the subscript the related functional comonomer in agreement with the notation “f” for the feed composition. “inst” is still used to denote instantaneous.

Specific comment 1

As <FUNC_{inst}> is equivalent to F_(A,inst), why not use F_{A,inst} throughout? Likewise, <FUNC> could be redesignated F_(A,cum), following standard polymer notation.

Answer to specific comment 1

We thank the reviewer for this suggestion (see answer comment above).

Specific comment 2

Xm is used throughout the text as an abbreviation for (total) monomer conversion - it would be easier to read if this were spelled out.

Answer to specific comment 2

Indeed to minimize confusion we can just write monomer conversion. The necessary changes have been made in the main text.

Specific comment 3

p2. "the synthesis of functional polymers is commonly performed..." this makes it sound as CROP and RDRP are the main techniques for functional polymer synthesis, it would be better to rephrase so that it is clear that these are two possibilities out of many.

Answer to specific comment 3

We agree with the reviewer. This has been altered in the introduction.

Specific comment 4

p6. I don't think x_n has been defined yet. Beware of possible confusion with X_m .

Answer to specific comment 4

Indeed it was only defined in the caption of Figure 2. This has been altered in the revised manuscript.

Specific comment 5

Figure 2 caption: 'most chains have a length of 100 and FUNC of 50' it would be more correct to say the most probable chain length and FUNC is 100 and 50 (i.e. this is the mode of the distribution).

Answer to specific comment 5

We have taken up this suggestion of the reviewer in the revised manuscript.

Specific comment 6

In Figure 3 and in Figures S1-S15, how was the size of the error bars determined? Do they represent the standard error, or a confidence interval? This should be specified.

Answer to specific comment 6

The reviewer is correct that this needs to be specified. The bars are based on the standard deviation. This has been highlighted in the experimental section.

Specific comment 7

The discrepancy between experimental and theoretical dispersities is real, and requires some comment - while the observed dispersities are broadly consistent with the predictions, the effect of temperature on dispersity that is predicted by the model is simply not observed. Perhaps this is due to the difficulty of obtaining accurate values, or perhaps the model is wrong - in any case, the discrepancy must be noted and some attempt should be made to explain it.

Answer to specific comment 7

In our previous reply letter, which was perhaps not accessible for the reviewer, we have highlighted the following paragraph: "It should be further stressed that a kinetic model – once based on a sufficient broad set of experimental data – can filter out unrealistic parameter combinations and

identify less reliable experimental data. Arrhenius behavior can be expected for individual reaction steps as correctly implemented in the kinetic model. As chain transfer is the contributor to the dispersity increase and the relevant importance of this reaction increases at higher temperature, in agreement with literature data, it is thus understandable why the optimized model predicts lower dispersities for the 100°C data in Figure 3d. This again illustrates the strength of a combined experimental and modeling approach.”

This discrepancy and the associated reasoning is also included just above Figure 3, hence, in our opinion this aspect has already been covered and no further changes are needed.

Specific comment 8

I don't think Figures S13 and S14 are necessary. The additional information provided by the figures is offset by the inconvenience of having to search for them in the Supporting Information. That the copolymer composition is different to the comonomer feed is well-understood, and I don't really see how it shows that the polymerization should not be stopped at lower conversion - frequently in this situation one runs the polymerization at a monomer feed that gives the desired copolymer composition, then stops the reaction before significant composition drift takes place (e.g. around 75% in this case). Also Fig S13 shows $\langle \text{FUNC} \rangle$ nearly always greater than $f_{\text{(C2MestOx)}}$, while Fig 4c shows $\langle \text{FUNC}_{\text{inst}} \rangle$ always $< f_{\text{(C2MestOx)}}$ - how is this possible?

Regarding Fig S14, it could simply be stated that an identical composition profile is obtained in the absence of chain transfer - this is not particularly surprising.

Answer to specific comment 8

We understand the comment of the reviewer. However, we are restricted in the number of figures and words to include in the main text, which automatically implies that at one point certain parts need to be covered in the Supporting Information. We decided to include this more technical part in the Supporting Information, as the main message (see next paragraph) is only relevant for the general reader. A specialized reader is then referred to the Supporting Information.

Regarding the interpretation of Figure S13 we understand the confusion of the reviewer as here the plot was somewhat unfortunately based on the non-functionalized monomer with a normalization based on the functionalized monomer. We thank the reviewer for noting this and in the revised version the updated plot has been added to solve this issue. By the introduction of the improved symbols, as also suggested by the reviewer (see above), such misinterpretations are also avoided as one always has the subscript now explicitly highlighting the monomer type.

The reviewer is correct that it is well-understood that the copolymer composition is different as in the feed composition. Also the statement regarding Figure S14 is as such expected if you are a more specialist in the field of mathematics and polymerization kinetics (as the reviewer for sure is). Typically one evaluates functionalization processes on average values and one ignores side reactions so that one ends up with a biased interpretation. For a general reader the illustration of these aspects are thus very relevant and therefore we opt to not remove this part from the overall manuscript. It is because of the side reactions that you for sure need to consider of FUNC-CLDs, which is not the traditional approach.

Regarding the statement on not stopping the polymerization earlier we would like to highlight that this has been attempted in the past by several research groups. However, the average profiles already show that only at the very high monomer conversions the functional comonomer is very prone to become incorporated. It is thus important to aim at very high monomer conversions here. To stress this more we have included that particular paragraph for the experimentalist.

Specific comment 9

On p12, the discussion of the origins of the trimodal fingerprint could refer to Figure 5 (e.g. the lower arm is due to chain transfer, the top arm to chain initiation after chain transfer). The reader could then be referred to Figure S15 (e.g. "this is illustrated in Figure S15, in which different side reactions are successively added to the polymerizations"). The idea is to avoid having to continually refer to the supporting information in order to understand the article.

Answer to specific comment 9

We understand the comment of the reviewer and have extended Figure 6 to avoid this switch to a part of Figure S15.

Specific comment 10

p14: "It should be stressed..." I don't think there's any need for this sentence, which appears meant for the reviewers/editor more than for the reader.

Answer to specific comment 10

We have rephrased this sentence as suggested by the reviewer.

Specific comment 11

Figure 6(c) looks wrong to me: the black distribution in Figure 6a roughly corresponds to a Poisson with mean 0.7, while the black Poisson distribution in 6c has a mean of roughly 1.5. But the means of the two distributions are stated to be the same. Maybe the authors could check this.

Answer to specific comment 11

We thank the review for making this point. We have updated the figure. In relation to this comment we have also checked the other figures and made updates when needed.

Specific comment 12

As Figure S27 duplicates Fig 7d, Fig S27 should be deleted, and all references in the text should be to Fig 7d. I still find the descriptions of functionalization quality entirely arbitrary and the characterization from "bad" to "excellent" seems unjustified. What is special about $< 2\%$ non-functional chains and $C_v < 0.4$? At the least, these criteria should be explained in the text (e.g. the text accompanying Fig S27 should be moved to the main article).

Answer to specific comment 12

We understand the comment of the reviewer. We have removed Figure S27 as Figure 7d suffices. The color mapping (from red to green, with intermediate colors of orange and yellow) is used to provide a guide of the eye for the reader. Upon a direct inspection of several colored FUNC-CLDs together the reader can directly see which systems behave better (green) and which don't (red). This is specifically useful if one compares reaction conditions as e.g. illustrated in Figure 8 with a lower row a higher target DP, and a higher column number a lower initial functional comonomer loading. The effect of both process variables can be directly observed, which improves in our opinion the accessibility for a general reader. The main message is the identification of trends based on a solid framework, as developed in the present work. A more specialist reader can then focus on the actual numbers associated with the color mapping (Figure 7d). Note that in our work the central values (transition toward green) are 0.05 for the fraction of non-functionalized chains and 0.5 for C_v (ratio of standard deviation and mean value). These numbers are arbitrary but as such quite logical, taking into account the general field of statistical analysis. 5% is often a threshold value for a distributed property in statistics. 0.5 is common for normalized standard deviations and intuitively easily accessible.

To emphasize this reasoning more we have adapted the section in which we introduce the color mapping:

“To enable a fast detection of the relation between reaction conditions and the functionalization quality the FUNCDCs, as obtained upon complete monomer consumption, are colored. The color code of Figure 7(d) is followed with a formal gradient from “bad” (dark red color) to “excellent” (dark green color). Intermediate colors are light green, yellow and orange. Note that the $C_{v,FUNCDC}$ and $f_{nonfunctionalized}$ boundaries defining these color changes are arbitrary but at least allow a ranking of functional copolymers impossible based on experimental research only. The key values (transition to green color; 0.5 for $C_{v,FUNCDC}$ and 0.05 for $f_{nonfunctionalized}$) are also common values used in the general field of statistics. Moreover, the color trends are relevant for application as purification by preparative SEC is impossible and preparative liquid chromatography for polymers is cumbersome.”

Specific comment 13

In the subsequent discussion, distributions are repeatedly described by their color or whether they are good or bad, and these arbitrary distinctions are thus invested with spurious meaning. Why is the difference between entries 9 and 10 of Table 9 significant, but the difference between entries 10 and 6 is not? I feel like these arbitrary categories make it more difficult to understand the results, not less.

Answer to specific comment 13

We understand the reasoning of the reviewer but as explained in the answer to the previous comment the main goal is to color distributions to enable a direct detection of trends. In a next step the focus can be on the actual numbers that have defined the trend. We do agree that the table is highlighting more numbers and therefore an alternative representation would be even better. This has been taken up in the revised manuscript, starting from the excellent suggestion of the reviewer later on (Specific comment 17).

Specific comment 14

On p18, the authors have inserted "the latter" - it's not clear if this refers to the too high C_v or the 'excellent' NON-FUNC.

Answer to specific comment 14

We thank the reviewer for noting this. This aspect has been covered in the revised manuscript.

Specific comment 15

On p19, it would be better to say "they are less likely to undergo chain transfer" instead of "they are more likely to not undergo chain transfer". The authors then state that the calculation of an average becomes less-defined for lower target DPs. The calculation of the average is still well-defined, perhaps they mean the concept of the average becomes less meaningful?

Answer to specific comment 15

We thank the reviewer for these suggestions. Indeed, these are better sentences.

Specific comment 16

On p21, it is great that the authors have been able to test their conclusions experimentally. I would note that if the number of functionalities per chain were Poisson distributed, one would expect $e^{-1.25} = 28.7\%$ of chains to be non-functionalized in this case - an agreement that is even better (more excellent?) than that of the model. On this point, I think it would be better to say that the experiment returned a result that was within 4% of the prediction, rather than using loaded terms like 'excellent'.

Answer to specific comment 16

We thank the reviewer for appreciating the experimental validation. We understand the comment of the reviewer regarding its success and have adapted the text part.

Specific comment 17

I have attached a graph that I suggest the authors use in place of Table 9, showing graphically the variation of C_v and NONFUNC as DP, f , monomer and T are varied. A graphical presentation makes the trends quite obvious, and erases the arbitrary excellent/very good/good/bad distinctions. Whatever the authors decide to do, I feel that the data should be discussed in terms of trends and effects of changing parameters, not in terms of whether or not they conform to an arbitrary goal.

Answer to specific comment 17

We would like to again thank the reviewer for his effort. Indeed the overview of our CROP work is further recognizable by including such a figure. We have replaced the table by the suggested figure but maintained in Supporting Information the original table in view of our explanation to the answer of specific comment 12. The main purpose of the colors is to provide a quick assessment of trends which is easy with a transition in color from green to red.

Specific comment 18

The statement on semi-batch polymerizations that has been added to p23 needs some context in order to be understood by a typical reader - why would semi-batch procedures be of interest?

Answer to specific comment 18

We understand the comment of the reviewer. We made the general reader more aware that in copolymerization processes often semibatch approaches are considered but in the current work such approaches are not the preferred route, taking into account our answer to one of the previous comments of reviewer 1: "The reviewer is correct that semibatch operation, which typically involves the addition of one of the comonomers with increasing time as opposed to the batch case with all comonomer present at the start, has been conducted in an industrial context. Essential is here the so-called Mayo-Lewis plot as also included in the present work (Figure 4). It should however be stressed that this relates to conventional chain growth polymerization (e.g. free radical polymerization) which does not allow the synthesis of the advanced well-defined polymer products as targeted in the present work. Here focus is on low dispersity well-defined products for advanced applications in the field of polymer therapeutics and hydrogels aiming at full monomer conversion, hence, batch conditions. Moreover, as illustrated in Figure 4, the Mayo-Lewis shape is rather unsuited for semibatch procedures and the key disturber for chain growth control is chain transfer."

General comment 3

In conclusion, I think this article could be accepted for publication, but still needs some fairly substantial revision, particularly concerning the discussion of functionalization quality. Removing the discussion of 'excellence' and replacing it with a simple explanation of what factors affect C_v and NONFUNC would make the article both shorter and better. In my opinion the value of this article lies less in presenting a detailed explanation of how to make a particular functional polymer to an arbitrary quality standard and more in its presentation of the FUNC-CLD distribution and how different parameters affect C_v and NONFUNC.

Answer to general comment 3

We thank the reviewer again for his general appreciation. As explained above we have improved the discussion regarding the interpretation of the functionalization quality. In particular, this has been significantly improved by also including the figure as suggested by the reviewer.